

# Analysis and Modelling of a 9.3 kyr Palaeoflood Record: Correlations, Clustering and Cycles

Annette Witt[1], Bruce D. Malamud[2], Clara Mangili[3], and Achim Brauer[3]

[1] Max Planck Institute for Dynamics and Self-Organisation, Göttingen, Germany
[2] Department of Geography, King's College London, London, U.K.
[3] GFZ German Research Centre for Geosciences, Potsdam, Germany

*Correspondence to*: Annette Witt (annette.witt@ds.mpg.de) and Bruce D. Malamud (bruce.malamud@kcl.ac.uk)

**Abstract:** In this paper we present a unique 9.5 m palaeo-lacustrine record of 771 palaeofloods which occurred over a period of 10 kyr in the Piànico-Sèllere basin (southern Alps) during an interglacial period in the Pleistocene (sometime 400−800 ka) and analyse its correlation, clustering and cyclicity properties. We first examine correlations, by applying power-spectral analysis and detrended fluctuation analysis (DFA) to a time series of the number of floods per year and find weak long-range persistence: power-spectral exponent $\beta_{PS} \approx 0.39$ and equivalent power-spectra exponent from DFA, $\beta_{DFA} \approx 0.25$. We then examine clustering using the one-point probability distribution of the inter-flood intervals and find that the occurrence of palaeofloods do cluster in time as they are Weibull distributed with shape parameter $k_W = 0.78$. We then examine cyclicity in the time series of number of palaeofloods per year, and find a period of about 2030 yr. Using these characterizations of the correlation, clustering and cyclicity in the original palaeoflood time series, we create a model consisting of the superposition of a fractional Gaussian noise (FGN) with a 2030 yr periodic component and then peaks over threshold (POT) applied. We use this POT$_{FGN+Period}$ model to create 2,600,00 synthetic realisations of the same length as our original palaeoflood time series, but with varying intensity of periodicity and persistence, and find optimized model parameters that are congruent with our original palaeoflood series. We create long realizations of our optimized palaeoflood model, and find a high temporal variability of the flood frequency, which can take values between 0 to >30 floods century$^{-1}$. Finally, we show the practical utility of our optimized model realisations to calculate the uncertainty of the forecasted number of floods per century with the number of floods in the preceding century. A key finding of our paper is that neither fractional noise behaviour, or cyclicity, are sufficient to model frequency fluctuations of our large and continuous palaeoflood record, but rather a model based on both fractional noise superimposed with a long-range periodicity is necessary.

**Key words:** varve, palaeoflood, time series analysis, event series, long-range correlations, clustering of events, cyclicity, long-range persistence, peaks over threshold, modelling



## 1 Introduction

Risk estimates of floods are often based on instrumental records that cover a comparable small time period (e.g., a maximum of 150 yr) and might be influenced by anthropogenic activities. Therefore, palaeoflood sequences represent a promising source for understanding the long-term hazard dynamics of the unperturbed climate system (e.g., Kochel and Baker, 1982; Stedinger and Cohn, 1986; Baker, 1987, 2006; Ely et al., 1993; O'Connor et al., 1994; Knox, 2000; Yang et al., 2000; Greenbaum et al., 2000; Redmond et al., 2002; Benito et al., 2004; Czymzik et al., 2010; Huang et al., 2013; Swierczynski et al., 2013; Wirth et al., 2013). One important aspect of these dynamics is to understand to what extent flood frequencies can be considered as constant over time, and more specifically, if floods in a given geographic region have any correlations with themselves and if the floods are clustered in time. Here we examine correlations, clustering and cycles in a 9336 yr record of palaeofloods using varved (annual) sediments from Palaeolake Piànico-Sèllere (northern Italy), and construct a model to take into account the observed behaviour. In this introduction we introduce the idea of *correlations* and *clustering* in time series, then summarize the overall organization of this paper.

We first consider *correlations* in a time series. Consider an uncorrelated time series (e.g., a white noise), where values are independent of one another, i.e., it is equally likely at each time step to have values above or below the median value. To illustrate correlations, take a flood intensity time series, with flood intensity the number of floods per year; a 'flood' here might be defined in many different ways. If the flood intensity time series is uncorrelated, when there is a year with a large intensity (a large number of floods occurring, above the median number of floods), it is equally probable to have the next year a number of floods that is above *or* below the median, i.e., a flood intensity value that is 'large' (more floods) *or* 'small' (zero or few floods). In contrast, for a flood intensity time series that exhibits positive correlations, adjacent values will have flood intensities that are on average closer to each other (in intensity) than for an uncorrelated time series; large values tend to follow large ones, and small follow small. Temporal correlations are also referred to as persistence or memory (see Witt and Malamud 2013, and references therein). An example of positive correlations in two time series are given in **Fig. 1A** and **1B**, using a tree ring standardized growth index for Bristlecone pine, White Mountain, California, USA, for the years AD 0–1962 and cosmic ray neutron counts per hour, Beijing, China, 1 January to 11 March 2008. In these two **Fig. 1** panels, successive values in both series are positively correlated with one another, and are examples of persistent time series.

Correlations can be both short-range (where only values in a time series close to each other are correlated) or long-range (where all values in the time series are correlated with one another). We will find that the palaeoflood time series used in this paper exhibit both long-range correlations and long period cyclical behaviour, and will focus on long-range (vs. short-range) persistent models in this paper. Long-range correlations have been discussed and documented for many processes in the Earth Sciences (Witt and Malamud 2013), with examples including river run-off and precipitation (Hurst, 1951; Mandelbrot and van Ness, 1968; Montanari et al., 1996; Koscielny-Bunde et al., 2006; Mudelsee, 2007; Khaliq et al., 2009, Ghil et al., 2011),





atmospheric variability (Govindan et al., 2004) and temperatures over short- to very-long time scales (Pelletier and Turcotte, 1997; Fraedrich and Blender, 2003). A more detailed description of correlations, particularly long-range correlations, will be given in **Sects. 3** and **4**.

An alternative to examining the temporal *correlations* of flood intensities (here taken as the number of floods per year), is to examine if the flood intensities over a given threshold *cluster* in time. An example of clustering, in unequally spaced wind speed strong storm/hurricane data over a given threshold, is given in **Fig. 1C** and **1D**. Here we observe the clustering on an annual cycle. Correlation and clustering (or lack of) have been studied for different extreme natural events, including earthquakes (e.g., Livina et al., 2005; Hainzl et al., 2006; Lennartz et al., 2008; Davidsen and Kwiatek, 2013), volcanic
eruptions (e.g., Nathenson, 2001), floods (e.g., Pelletier and Turcotte, 1997; Milly and Wetherald, 2002), and tropical temperatures (e.g., Blender et al., 2008).

In this paper, we use a record of 771 detrital layers of the laminated sediments of the Palaeolake Piànico-Sèllere (northern Italy) covering a time span of 9336 yr. These 771 layers are interpreted as indictors of flood events which occurred somewhere
during the period of 393 to 780 ka (thousands of years ago). In **Sect. 2** we give a detailed description of the palaeoflood sequence. Then in **Sect. 3** we analyse the distribution of the palaeoflood frequencies per year and per decade, the temporal correlations among the flood events and the one-point probability distribution of the inter-flood intervals. In **Sect. 3** we also model the record of palaeoflood timings as elements of a long-range correlated synthetic time series that exceed a threshold. For this model type, scaling relationships for the distribution of the inter-event intervals and the temporal correlations are
derived in dependence on correlation properties of the time series and on the applied threshold. In **Sect. 4** we construct a model that consists of multiple realisations of synthetic time series that captures the same clustering and correlation properties that we find in our palaeoflood record. Finally, in **Sect. 5** we provide a summary of the paper.

## 2 Data

As discussed in the introduction, this paper focuses on the correlation, clustering and cyclicity of palaeofloods. We present
here for the first time and use a very comprehensive flood record at sub-annual resolution, obtained from the varved sediments of the Piànico-Sèllere Basin, located in the Borlezza Valley (Province of Bergamo, Italy; **Fig. 2A**). This palaeolake sequence was first described in the mid 1800s (e.g., Stoppani, 1857), mainly for its fossil content. A detailed stratigraphic study was published by Moscariello et al. (2000).

The Palaeolake Piànico-Sèllere is located at the foothills of the Southern Alps in Italy. Its sediments (45°48′ N, 10°2′ E, 280−350 m above sea level) are visible in outcrops (**Fig. 2B**). The sediment formation is more than 48 m thick and extends for about 600 m laterally. It includes four fine-grained laminated stratigraphic units (described more fully below). The size of the



palaeolake has been reconstructed to about 3 km in length and 500−800 m in width (Casati, 1968). The palaeocatchment area of the Piànico-Sèllere Lake was < 13 km² (Moscariello et al., 2000).

The lacustrine sequence that forms the Piànico Formation (Moscariello et al., 2000), is a 48 m thick stratigraphic interval that
includes four units. For the palaeoflood dataset created here, only the 9.5 m unit called BVC (*Banco Varvato Carbonatico* or Carbonate Varved Bed) will be considered (described more fully below). The age of the sediment is still debated. Tephrochronological dating of the sequence gives 393±12 ka (Brauer et al., 2007), which corresponds to the interglacial period at about 400 ka, i.e. the Marine Isotope Stage (MIS) 11. This interglacial period is considered as the best analogue to the Holocene because of similar orbital parameters (e.g., Loutre and Berger, 2003). However, Pinti et al. (2001) dated a tephra
layer in the varved BVC sequence at 779±13 ka, assigning the sequence to the interglacial at about 780 ka (MIS 19). For the purposes of the paper here, we will be less concerned with the actual age, and more concerned about the period of overall time that has elapsed, based on interpreting the BVC alternating layers (rhythmites) as varves.

The BVC 9.5 m unit constitutes an almost continuous succession of about 15,500 rhythmites (alternating layers of dark and
light sediment), 0.2−0.8 mm thick (Moscariello et al., 2000). These endogenic calcite rhythmites have been interpreted to be varves (annual cycles) as they have a structure very similar to Holocene Alpine lake varves, in which calcite precipitation take place in spring and summer (Kelts and Hsü, 1978; Moscariello et al., 2000). The Piànico-Sèllere varved sequence formed under interglacial conditions, as testified by the flora remains included in the sediments (e.g., Amsler, 1900; Maffei, 1924; Rossi, 2003; Martinetto, 2009), and the oxygen stable isotope values of this interval (Mangili et al., 2007). The calcite varves
consist of two layers (**Fig. 2B**): a lightly coloured and ~0.5 mm thick spring/summer layer formed by up to 96% of endogenic calcite, and a dark and thin winter layer constituted of organic remains, diatom frustules and occasional detrital grains (Moscariello et al., 2000; Mangili et al., 2005). Using wavelet analysis, Brauer et al. (2008) found that the varve thickness is partially modulated by solar activity (88 yr and 208 yr cycles), and probably also by the thermohaline ocean circulation (512 yr cycle). In the BVC stratigraphic interval, the upper 60% (9271 out of 15,500 varves) were examined here in detail using the
same methodology and location as described in Mangili et al. (2005), but extending the vertical profile from 896 varves (Mangili et al., 2005) to 9271 varves (here).

We briefly describe our microfacies sampling methodology (see Mangili et al., 2005 for further details). Continuous vertical profiles of sediment samples were collected from two outcrops stratigraphically similar, 150 m apart: (i) the *Main Section*, for
which data is presented in this paper, and (ii) the *Wall Section*, a control section which we used to evaluate uncertainties in the Main Section results. Both sections had more than 30 key marker layers so the two outcrops could be correlated with each other. For each vertical section, the outcrops were cleaned with a sharp knife until a smooth and vertical surface was obtained. Then a block of sediment was carved out *in situ* to enable easy pushing of a special stainless box (33 cm x 5 cm) with removable side walls onto the sample. Samples were taken with at least a 5 cm vertical overlap, ensuring that a marker layer, allowing



correlation, was present in both samples. The samples were then slowly dried at room temperature to avoid shrinkage and cracking and covered with a transparent epoxy resin, resulting in resin impregnation of the surface layer (1–2 mm) of the sediments. Samples were cut into two halves with the fresh surface again carefully dried and impregnated. From one half, 10 cm thick samples with a 4 cm overlap for final thin-section (120 mm x 35 mm) preparation were cut out. The thin sections

were analysed with a petrographic microscope. For measurement of varve and detrital layer thickness, 100x magnification was chosen.

Approximately 8% of the varve couplets contain also one or two detrital layers (**Fig. 2B**). The detritus is mainly constituted of Triassic dolomite from the catchment (Mangili et al., 2005). Due to their composition and grain size, these detrital layers can

easily be distinguished from the background endogenic sedimentation of the lake. These detrital layers are considered to be the result of channelized streamflow that originated in the hills surrounding the Piànico-Sèllere Basin and triggered by extreme precipitation events (Mangili et al., 2005). The position of a detrital layer within a varve allows the identification of the season in which the extreme precipitation event/flood took place: a 'spring' detrital layer settled before the beginning of endogenic calcite precipitation, a 'summer' detrital layer is within the varve summer layer and a 'fall/winter' detrital layer is at the top of

the calcite layer or included in the winter layer (Mangili et al., 2005). Due to the reduced thickness of the varve winter layer (0.06 mm mean), fall and winter detrital layers are not distinguished here. Another layer type (called matrix-supported) has also been observed in the varved sediments (Mangili et al., 2005); these layers are thought to result from reworking processes within the lake, are not linked to extreme precipitation events and will, therefore, not be taken into consideration in this paper.

In our analysis, we will focus on the temporal succession of detrital layers (flood events), the series of events that is graphically presented in **Fig. 2C**, and which consists of 771 single palaeoflood events occurring during a time interval of $T = 9336$ yr. There are 9271 varves present, each taken to represent 1 yr, in addition to a 'gap' of 65 yr (varve index 4019−4083) which is due to a slump at the Main Section. The varve structure there was not recognizable. To ascertain the temporal period of the gap, we correlated three marker layers between the Main Section's disturbed interval, to an undisturbed sequence at the Wall

Section, where we counted the number of varves, and were able to conclude that the gap in the Main Section is 65 yr. The flood events have each been given a specific varve index (relative age) in the time series. In this paper, we will use the following notation (see also **Table 1** for list of all abbreviations and notation used in this paper) for the observed data with respect to varves and detrital (flood) layers: $n_{year}[t]$, $t = 1, …, N_{varves}$ is the number of detrital layers per varve (flood events per year) where the index $t$ is the varve number (starting from the top varve) and represents relative age in years (see above) and $N_{varves}$

$= 9336$ yr is the length of the observational period, i.e. the total number of varves including the 'gap' of 65 yr.

The thicknesses of the detrital layers representing the 771 flood events that we investigate in this paper range from 0.02 to 23.00 mm (with quartiles of 25%: 0.07 mm, 50%: 0.10 mm and 75%: 0.18 mm) (see **Appendix A, Fig. A1**). Although detrital layers indicate the occurrence (year) of an extreme precipitation event (flood), the thickness is not taken as representative of





the magnitude of the event, as the thickness of a layer may depend on the distance between the source of the detritus and the studied section, the amount of detritus available at the time of the extreme precipitation and weaker vs. stronger rainfall during the extreme precipitation events. In addition, over the 9336 yr period we investigate, the source of given storms might result in different thickness signatures. We also note that although we can identify when a flood is thought to have occurred, this

does not preclude the possibility that other floods occurred but were not identified as a detrital layer. We do acknowledge research (e.g., Corella et al., 2014; Schillereff et al., 2014) where the type and thickness of detritus in Holocene varves are used to make interpretations of the strength of a hydro-meteorological event; however, due to the age of our sediments this sort of analysis of event magnitude was not possible here. In this paper, we therefore focus on just the temporal attributes of the 771 floods themselves over this 9336 yr period in the Pleistocene. See data availability for access to the palaeoflood

database.

In terms of the relative temporal uncertainty of the floods (based on the varve sequence), a flood in a given year (represented by the detrital layers) can (very rarely) erode some of the sediment underneath it, forming a hiatus of missing sediment. This limitation is discussed in detail in Mangili et al. (2005), who analysed for 10% of the outcrop presented in this paper, the same

stratigraphic intervals in both the Main Section and the Wall Section, again 150 m apart. Out of 896 varves they examined in both sections, they found that 3 varves out of 896 were not present (e.g., eroded away by a flood) in the Wall Section but were present in the Main Section, and 4 varves out of 896 were not present in the Main Section, but present at the Wall Section. Overall, this represented a total of 889 out of 896 varves (99.2%) in common between the two outcrops, which means that if we extend their results to the other 90% of the outcrop, some uncertainty might exist with up to 1% of the varves 'missing' in

time, which should have a very small impact on the relative timing of floods discussed in this paper.

The analyses in this paper will concentrate on the fluctuations in flood frequency over time (i.e., the number of floods per year and per decade), examining the temporal *correlations*, *clustering* and *cyclicity* of the flood frequencies.

### 3 Statistical metrics, correlations and clustering of the palaeoflood sequence

In this section we first statistically analyse the 771 timings of palaeofloods in terms of their elementary metrics of number of palaeofloods per year, decade and century compared to a Poisson process (Sect. 3.1). We then characterise the temporal clustering of the palaeoflood time series using autocorrelation (Sect. 3.2) and interevent occurrence time (Sect. 3.3). Next we characterize the temporal correlations of the time series using detrended fluctuation analysis and power-spectral analysis (Sect. 3.4). Finally, we characterize the cyclicity of the palaeoflood time series using a sinusoidal model (Sect. 3.5). As the techniques

we used are designed for time series with a continuous one-points probability distribution (e.g., Gaussian, log-normal or Levy distribution) vs. the integer values that are found in our palaeoflood time series, we throughout also show the statistical significance of our findings.



### 3.1 Number of palaeofloods per year, decade and century, compared to a Poisson process

The timings of the considered 771 palaeoflood events are transformed into three event series which will be analysed in detail in the subsections of this **Sect. 3**. We will work here with of the number of floods (detrital layers) per year ($n_{year}$), per decade ($n_{decade}$) and per century ($n_{century}$), over the 9271 yr of palaeoflood data (over a 9336 yr record) which are presented in **Fig. 3A**.

These data sets are integer-valued time series which contain $N_{year} = 9271$, $N_{decade} = 925$ and $N_{century} = 92$ data points.

The histograms of these data are given in **Fig. 3B**. The number of years ($k_{year}$) with no floods ($n_{year} = 0$ floods $yr^{-1}$) is $k_{year} = 8530$ yr (92.0% of the record). Similarly, for $n_{year} = 1, 2, 3$ floods $yr^{-1}$, the number of years (respectively) with those values are $k_{year} = 712$ (7.7%), 28 (0.3%), 1 (0.01%) yr. For $N_{year} = 9271$ yr examined, there are 741 yr which have 1 to 3 floods (detrital

layers), a total of 771 floods. We will revisit these values below. The data cover $N_{decade} = 925$ non-overlapping decades (because of the 65 yr gap, eight of the decades within the 9336 yr are disrupted). The number of decades ($k_{decade}$) with no floods ($n_{decade} = 0$ floods $decade^{-1}$) is $k_{decade} = 53$ decades (49.0% of the record). Similarly, for $n_{decade} = 1, 2, 3$ floods $decade^{-1}$, $k_{decade} = 283$ (30.6%) 123 (13.3%), 43 (4.6%) decades. For $4 \leq n_{decade} \leq 8$ floods $decade^{-1}$, $k_{decade} = 23$ (2.5%) decades. No decades have $n_{decade} > 8$ floods $decade^{-1}$. If we consider the data on the century resolution, we have $N_{century} = 92$ non-overlapping centuries,

as the 65 yr gap falls completely within 1 century. The number of centuries ($k_{century}$) with no floods ($n_{century} = 0$ floods $century^{-1}$) is $k_{century} = 1$ century (1% of the record). For $1 \leq n_{century} \leq 5$ and $6 \leq n_{century} \leq 10$ floods $century^{-1}$, $k_{century} = 26$ and 35 centuries (28% and 38% of the record) respectively. For $n_{century} > 10$ floods $century^{-1}$, $k_{century}$ then decreases (**Fig. 3B**) with a maximum of one century that has $n_{century} = 31$ floods $century^{-1}$.

There are time periods with little fluctuation of the number of floods per century (i.e., the interval from the 18[th] to the 24[th] century with values between 10 and 16 floods per century). However, we also see sudden transitions from a few to many floods (as from 4 to 24 floods from the 38[th] to the 39[th] century) as well as from many to a few floods (as from 25 to 7 floods from the 44[th] to the 45[th] century).

We now examine the number of floods per year, decade and century, as if the process that created them resulted in a series of floods uncorrelated in time. Here, we will consider a time series to be the realisation of the underlying process. If we are able to show that our time series has properties different from a Poisson process, we can infer that it is correlated or clustered and therefore non-stationary. Poisson processes have been used to model flood occurrences (e.g., Kirby, 1969; Todorovic and Zelenhasic, 1970). The 'validity' of the Poisson distribution has been discussed by various authors (e.g., Cunnane, 1979) and

more complex Poisson-cluster processes have been applied for modelling rainfall. Other authors have shown that Poisson processes are inappropriate, for example Mudelsee et al. (2004) who showed that the winter floods of the river Elbe from 1500 – 2000 cannot be modelled by a stationary Poisson process.



We will use as a model a Poisson process with a constant rate parameter to create model time series of flood events per year, decade and century (i.e., similar to the palaeoflood time series shown in **Fig. 3A**). We will see below that the resultant model realizations will be stationary, uncorrelated in time and non-clustered, and then show in subsequent sections that this modelling is inappropriate for our palaeoflood time series. For a time series to be considered a realisation of a Poisson process the following must be true (Cox and Lewis, 1978):

(i)     the time series elements are non-negative integers;

(ii)    the time series considered is stochastic (i.e., has no obvious trends and periodicities and no deterministic components);

(iii)   the one-point probability distribution of the time series elements follows a Poisson distribution, that is a positive integer-valued random variable $X$, whose probability densities $P_\lambda$ are defined as (Cox and Lewis, 1978):

$$P_\lambda \left( X = n \right) = \frac{\lambda^n}{n!} \exp\left(-\lambda\right), \quad n = 1, 2, 3, \ldots \tag{1}$$

where '!' means factorial, exp is the exponential function, $\lambda$ is the rate parameter and $n$ is the intensity (number of occurrences) of the 'random' variable per a given time unit (e.g., hour, year, decade, century). The rate parameter $\lambda$ is a physical quantity and thus has units (time unit)$^{-1}$, but the Poisson process as given in **Eq. (1)** is a mathematical model and $P_\lambda$ does not have units.

(iv)    the times between successive events (interevent occurrence times, IEOTs) are independent in time (i.e., there is no correlation between one IEOT and others, so events occur independent of one another);

(v)     the frequency-size distribution of the IEOTs is exponential.

We first interpret the number $n$ in **Eq. (1)** as the number of detrital layers per varve ($n_{\text{year}}$, floods yr$^{-1}$). The rate parameter $\lambda$ controls the probability of an event: the mean number of events per year (i.e., per varve) is $\overline{n}_{\text{year}} = \lambda_{\text{year}}$ and the variance of the number of events is $\sigma^2\left(n_{\text{year}}\right) = \lambda_{\text{year}}$. This equation holds only mathematically but not physically, i.e., it holds for the numbers but not for the units. The probability in **Eq. (1)** is not time-dependent, and rather only depends on (if we take our unit of time to be one year) the probability of $n_{\text{year}} = 1$ flood yr$^{-1}$, $n_{\text{year}} = 2$ floods yr$^{-1}$, etc., and not the relative temporal spacing of the floods in time. For modelling the series of the number of flood events per year in the 9.3 kyr Piànico-Sèllere palaeoflood time series, we use a (constant) rate parameter of $\lambda = 0.083$ floods yr$^{-1}$ that is the mean number of detrital layers per varve (i.e., (771 floods)/(9271 yr) = 0.083 floods yr$^{-1}$, where we discard the 65 yr gap for the total number of years considered).

Another consequence of considering the number of floods per year as a Poisson process is that the time interval $\Delta$ between two successive events (i.e., the interevent occurrence times (IEOTs) between two successive floods) follow an exponential one-point probability distribution $P$ for very small time units (Cox and Lewis, 1978):

$$P_{\text{exp}}\left(\Delta\right) = \lambda \exp\left(-\lambda\Delta\right), \quad \Delta \geq 0 \tag{2}$$



with the same rate parameter $\lambda$ as in **Eq. (1)**. Now, the relative ordering between single floods is taken into account. For example, if in a given decade, we have three varves each with one flood, **Eq. (2)** will be different if the three floods are the first three years of the decade (i.e., mean $\Delta = 1$ yr) vs. a flood in years 1, 5, and 9 of the decade (i.e., mean $\Delta = 4$ yr). Note, that the time interval $\Delta$ can have non-integer values, for instance if two or more events (floods) occur in a single time unit (within one year).

For our palaeoflood time series (**Fig. 3A**), we now consider whether the histograms (vertical bars in **Fig. 3B**) of the observed number of years, decades and centuries with a given number of floods per year, decade and century follow Poisson distributions. For each palaeoflood time series, we created 100 realisations of a Poisson process, each realisation with 9271, 927 and 92 (respectively, for year, decade and century) time series elements and with rate parameters $\lambda = \lambda_{year} = 0.083$ floods yr$^{-1}$ = $\lambda_{decade} = 0.83$ floods decade$^{-1}$ = $\lambda_{century} = 8.3$ floods century$^{-1}$ (the measurements values are different as well as the units, the differences cancel out because of multiplication of measurement values and units). We discard the 65 yr gap for the purposes of this model. Each realisation therefore has values that are uncorrelated in time and follow a Poisson distribution (**Eq. 1**). Many common software programmes (e.g., Excel, R, Matlab) are able to easily generate such realisations.

We give the results of our Poisson model for each respective time resolution (year, decade, century) in **Fig. 3B** as diamonds (mean of the 100 realisations) ± error bars (standard deviation of the 100 realisations). We find that for the yearly data, the number of years observed ($k_{year}$) with a given number of floods per year, $n_{year} = 0, 1, 2, 3$ floods yr$^{-1}$, follows closely the number of years given by the Poisson model. For example, returning to the values given above, the number of years with $n_{year} = 0, 1, 2, 3$ floods yr$^{-1}$ is observed to be $k_{year} = 8530, 712, 28, 1$ yr (respectively), and with the Poisson model $k_{year} = 8531 \pm 26, 709 \pm 25, 30 \pm 5, 1 \pm 1$ yr (mean ± standard deviation) (respectively). In contrast, for both the decadal and centennial data, the observed data set contains decades and centuries with many fewer or greater floods compared to the model data (see **Fig. 3B**). For example, the number of decades with $n_{decade} = 0, 1, 2$ floods decade$^{-1}$ is observed to be $k_{decade} = 453, 283, 123$ decades (respectively) with the Poisson model giving $k_{decade} = 406 \pm 15, 338 \pm 13, 140 \pm 11$ decades (mean ± standard deviation) (respectively). We therefore conclude that at the yearly scale the observed data follow a Poisson process, but at the decadal and centennial scales, the time series are more clustered than realisations of a Poisson process. This clustering property will be qualitatively and quantitatively characterized in **Sects. 3.2 and 3.3**.

### 3.2 Temporal correlations using autocorrelation

There are a variety of methods that can be used to explore temporal correlations. Correlations (also called persistence, memory) have been studied in many environmental time series (e.g., see Box et al., 2013 and Witt and Malamud, 2013 for reviews). Measures for correlations quantify the statistical dependence between variables, and in particular, measures for *temporal* correlations describe the intensity of this relation between time series elements with a fixed temporal distance. To quantify the





strength of correlations, we use autocorrelation analysis applied to the time series of number of floods per year ($n_{year}$), decade ($n_{decade}$), and century ($n_{century}$) (**Fig. 3A**). The autocorrelation function $C$ (e.g., Priestley, 1982) for the number of floods is defined as:

$$C(\tau) = \frac{1}{N\sigma^2(n)} \sum_{j=1}^{N-\tau} (n_j - \overline{n})(n_{j+\tau} - \overline{n}) \tag{3}$$

where $n_j$ is a time series of the number of floods per year (decade, century) with $j = 1,2,\ldots,N$, with $N = N_{year}$, $N_{decade}$ or $N_{century}$ (i.e., the total number of years, decades, centuries in the record), $\overline{n}$ is the mean number of floods (per year, decade, century) for the entire record, $\tau$ is the time lag (in years, decades, centuries), and $\sigma^2(n)$ is the variance of the time series. Positive values of the autocorrelation function are indicative of a process that on average, for a given lag $\tau$, has positive persistence between values separated by that lag $\tau$. In other words, if there are a large number of floods in a given year, on average, $\tau$ years later

will also be followed by a large number of floods (compared to the mean number of floods per year overall). The converse is also true, if there are few floods in a given year, on average, $\tau$ years later will also be followed by only a few floods (compared to the mean number of floods per year overall). Negative values of the autocorrelation function are indicative of negative persistence, whereby if there are a large number of floods in a given year, on average, $\tau$ years later will be followed by a very few number of floods (compared to the mean number of floods per year overall).

In our analyses of the palaeoflood sequence, only pairs of years, decades or centuries $(n_j, n_{j+\tau})$ which do not contain the 65 yr gap in the 9336 yr sequence of floods (detrital layers) are considered. The autocorrelation function applied to the times series of the number of floods per year, decade and century are given as correlograms in **Fig. 3C** (squares). This correlogram of the yearly floods shows significant positive correlations with a decaying trend, and no obvious periodic components for lags

of $1 \leq \tau \leq 200$ yr. Due to the strongly fluctuating behaviour of the annual data's correlogram (squares) in **Fig. 3C** it is difficult to determine the functional form of the decay, i.e., to decide if the decay is exponential (thin tailed, short-range persistent) or approaching a power-law (heavy tailed, long-range persistent). Therefore, we apply the autocorrelation function to the time series of the number of floods per decade (with respect to non-overlapping ten-year time intervals) and find (**Fig. 3C**) a power-law behaviour

$$C(\tau) \sim \tau^{-\gamma_{decade}}, \ \tau > 0 \tag{4}$$

with a power-law exponent of $\gamma_{decade} = 0.34 \pm 0.04$ (fitted for lags $1 \leq \tau \leq 20$ decade, which is equivalent to $10$ yr $\leq \tau \leq 200$ yr) with uncertainties $\pm 1$ standard error of the exponent. We will return to this power-law model (**Eq. 4**) in **Sects. 3.5 and 4.2**. We

conclude that for the floods per decade time series, when using the autocorrelation function, the data exhibit long-range correlations.



The time series of the number of floods per century contains only 92 data points. We therefore have calculated the autocorrelation function $C$ only for lags $\tau = 1$ century and $\tau = 2$ century as graphically presented in **Fig. 3C**. Based on this figure, we see significant positive correlations and thus confirm the results found for the temporal correlations of the number

of floods per year and decade. Due to the very few lag values, however, we cannot discuss the functional shape of this autocorrelation function.

We conclude that the three considered time series $n_{year}$, $n_{decade}$ and $n_{century}$ have positive autocorrelations for lags of $\tau < 200$ yr. For the number of floods per decade, $n_{decade}$, we find indications of a power-law shape of the autocorrelation function which

indicates long-range persistence of the time series.

### 3.3 Interevent occurrence times (IEOTs)

Here, we introduce the series of interevent occurrence times of the palaeoflood record which are derived as follows ($t$ is the relative age in years in our time series and as previously, $n_{year}$ is the number of floods per year).

- First, we define a *flood year* to be any year $t$ with at least one flood in it, i.e. $n_{year}[t] \geq 1$ flood $yr^{-1}$.

- Second, we define IEOT as the time interval $\Delta$ between successive years that have 1 or more floods, i.e., more formally, $n_{year}[t] \geq 1$ flood $yr^{-1}$ and $n_{year}[t + \Delta] \geq 1$ flood $yr^{-1}$ but for all years in-between, $n_{year}[t + l] = 0$ floods $yr^{-1}$ for $l = 1, 2, …, \Delta–1$.

In **Fig. 4A** we illustrate finding the IEOTs: we present the number of detrital layers per year (**Fig. 4A.1**) with as inset (**Fig.**

**4A.2**) an example for four detrital layers with IEOTs of $\Delta = 52, 3$ and $26$ yr. In **Fig. 4B** we give all the $\Delta$ as a function of relative age $t$ for the entire palaeoflood data set. The values of $\Delta$ range from 1 to 125 yr (with quartiles of 25%: 3 yr, 50%: 7 yr, and 75%: 15 yr) and a mean $\overline{\Delta} = 12.5$ yr. We then use a subscript $j$ to indicate successive IEOTs, $\Delta_j$, which we plot using 'natural' time, in other words where each $\Delta$ is no longer represented temporally when it occurs, but rather successively one after another, $j = 1, 2, 3, ..., N_\Delta$ with the total number of natural time intervals $N_\Delta = 739$ (we discard the $j$ value corresponding

to the gap). The use of natural time (i.e., the time between all events is 'equal' in natural time) to represent unequally spaced events in time series has been mainly used to examine seismic related time series (e.g., Uyeda et al., 2009; Rundle et al., 2012), but has found use in a variety of other disciplines ranging from biology and environmental sciences to cardiology (see Varotsos et al., 2011 for a review of its use in various disciplines). In **Fig. 4C** we plot the same $\Delta$ from **Fig. 4B**, but now as a function of detrital layer index (natural time), $j = 1, 2, 3, …, N_\Delta$.



### 3.3.1 One-point probability density of interevent occurrence times (IEOTs)

Here, we analyse the one-point probability distributions of the interevent occurrence times (IEOTs). As discussed in **Sect. 3.1**, a realization of a Poisson process is characterized by 'events' that are uncorrelated with one another and the $\Delta_j$ have an exponential distribution. Conversely, it means that if the $\Delta_j$ (the IEOTs) are non-exponentially distributed, the process is non-Poisson. In **Fig. 5A**, we give the frequency density of $\Delta$ (the one IEOT that includes the sediment gap is excluded) of the flood record along with an exponential distribution which represents the interevent occurrence times of events in a Poisson process. The distribution of the IEOTs has a higher number of both short and long time intervals compared to an exponential distribution. This observation is supported by statistical hypothesis testing where an Anderson-Darling test (Anderson and Darling, 1952) was adjusted to integer values of the IEOT distribution (using a method described in Choulakian et al., 1994; Arnold and Emerson, 2011), and the test rejected the null hypothesis that this distribution can be explained as the result of a Poisson process ($p$-value $< 0.002$).

We have shown above that our data are not a Poisson process, and now wish to further characterize the distribution of the IEOT. If the $\Delta_j$ (the IEOTs), in addition to not being exponentially distributed, are Weibull distributed, this is taken as an indicator for clustering of the original time series (Bunde et al., 2005; Witt et al., 2010). The two-parameter Weibull probability distribution (Weibull, 1951) is a standard waiting time distribution, i.e., frequently used for modelling the time intervals between successive events (Cox and Lewis, 1978). The continuous (vs. discrete, as in **Eq. (1)** for the Poisson distribution) two-parameter Weibull probability distribution given by (Weibull, 1951):

$$P_W\left(\Delta\right) = \frac{k_W}{\lambda_W}\left(\frac{\Delta}{\lambda_W}\right)^{k_W - 1} \exp\left(-\left(\frac{\Delta}{\lambda_W}\right)^{k_W}\right), \Delta \geq 0 \tag{5}$$

where the two parameters are $\lambda_W$ for scale and $k_W$ for shape, and $\Delta$ is any real number $\geq 0$. When the shape parameter $k_W = 1.0$, the two-parameter continuous Weibull distribution (**Eq. 5**) turns into:

$$P_W\left(\Delta\right) = \frac{1}{\lambda_W}\exp\left(-\frac{\Delta}{\lambda_W}\right), \Delta \geq 0, \tag{6}$$

that is the exponential distribution which describes the IEOT distribution of a Poisson process with a parameter $\lambda = 1/\lambda_W$ (see **Eq. 1**).





For shape parameters $0.0 < k_W < 1.0$, the two-parameter Weibull distribution (**Eq. 5**) is heavy-tailed (i.e., asymptotically scale invariant) with a tail parameter of $(1 - k_W)$ which means that the probability density for very large values $\Delta$ in **Eq. (5)** scales with a power law with the tail parameter as power-law exponent. For the palaeoflood IEOTs, we use a maximum-likelihood estimation (MLE) and assume a two-parameter Weibull distribution (**Eq. 5**). We find (see **Fig. 5A**) a best fit shape parameter for the Weibull distribution of $k_W = 0.91 \pm 0.024$. However, this result is questionable as the MLE technique is tailored to a Weibull distribution as a *continuous* probability distribution, but the interevent occurrence times $\Delta$ have discrete (integer) values. Therefore, we use benchmarks (500 realisations of a Weibull distributed model for random numbers) for fitting Weibull distributions to integer valued data. We find that the MLE estimator tends to over-fit the shape parameter by 0.12, i.e., we can correct our estimate of the shape parameter to $k_W = 0.79 \pm 0.024$.

### 3.3.2 Auto-correlation of interevent occurrence times (IEOTs)

Another indicator for clustering is positive temporal correlations of the interevent occurrence times. Such correlations are particularly caused by a 'lumping' of small (large) IEOTs which results in time intervals with an increased (decreased) flood rate which correspond to flood clusters (phases of little flooding activity). In **Fig. 5B**, we examine correlations (vs. clustering) of the IEOTs by applying autocorrelation analysis to the $\Delta_j$ for the IEOTs. We find significant positive correlations of the $\Delta_j$ for flood year lags $1 \leq \tau_\Delta \leq 20$ (no units). This indicates that for the IEOT, small values of $\Delta_j$ tend to follow small ones, and large ones tend to follow large. The correlogram (see **Fig. 5B**) of $\Delta_j$ exhibits a power-law behaviour

$$C\left(\tau_\Delta\right) \sim \tau_\Delta^{-\gamma_\Delta}, \gamma_\Delta > 0 \tag{7}$$

with a power-law exponent of $\gamma_\Delta = 0.45 \pm 0.14$ (fit for lags of $1 \leq \tau_\Delta \leq 20$ IEOTs, which is equivalent to $12.5 \text{ yr} \leq \tau \leq 250 \text{ yr}$). Power-law correlations for IEOTs are reported for a class of theoretical models by Bunde et al. (2003) and Eichner et al. (2007) who have studied peaks over thresholds of long-range correlated time series. We will explore this in more detail below.

In summary, we have analysed the one-point probability distribution and the autocorrelation function of the IEOTs of the palaeofloods with respect to a time resolution of 1 yr (i.e., we have not used time resolutions between palaeofloods that are sub-annual, such as three months). The distribution of the IEOTs can be well approximated by a Weibull distribution with a shape parameter of $k_W = 0.78$. Therefore, the IEOTs are more likely to be very short or very long temporal periods compared to a random (uncorrelated) occurrence of the IEOTs and therefore the palaeoflood time series is not a realisation of a Poisson process. Furthermore, the IEOTs have positive temporal correlations. This is particularly caused by a clustering of the very high and very low IEOTs and thus by a temporal clustering of the floods. The autocorrelation function seems to follow a power-law behaviour.



### 3.4 Quantifying long-range correlations using DFA and power-spectral analysis

We now return to correlations so that we can quantify the degree of correlations in our data, at yearly, decadal, and centennial scales. In this section we will start with a general statistical background on long-range correlations and discuss long-range versus long-range correlations, then we describe the idea of long-range persistence and fractional noises, and then use these ideas to quantify our data.

### 3.4.1 Definitions and background

Temporal correlations describe the statistical dependence of directly and distantly neighboured values in a time series. If the correlations are essentially linear and thus can be described by the autocorrelation function, two types of correlations can be considered: (i) short-range correlations (Priestley, 1982; Box et al., 2013) and (ii) long-range correlations (e.g., Taqqu and Samorodnitsky; 1992; Beran, 1994; Malamud and Turcotte, 1999).

Short-range correlations are characterized by a decay of the autocorrelation function $C(\tau)$ (**Eq. 3**) that is bounded by an exponential decay for large lags, $\tau$:

$$\left| C\left(\tau\right) \right| \leq \gamma_0 \exp\left(-\gamma\tau\right), \quad \tau > \tau_0 , \tag{8}$$

where $\tau_0$, $\gamma_0$ and $\gamma$ are non-negative constants. In particular, this definition applies for time series with a finite correlation length ($C(\tau_0) = 0$ for $\tau > \tau_0$). Statistical models for short-range correlated time series include autoregressive (AR) and moving average (MA) processes (Priestley, 1982).

In contrast to short-range correlated time series, long-range correlated time series approach a power-law decay of the autocorrelation function $C(\tau)$ (**Eq. 3**) for large lags $\tau$:

$$C\left(\tau\right) \sim \tau^{-(1-\beta)}, \, 0.0 \leq \beta \leq 1.0 \tag{9}$$

The parameter $\beta$ ($0.0 \leq \beta \leq 1.0$) is called the strength of the long-range correlations, with $\beta$ the strength of long-range correlations. The limiting values (for the autocorrelation function) $\beta = 0.0$ represents short-range persistence (see **Eq. 8**) between the time series elements, and $\beta = 1.0$ for pink (or $1/f$) noises. Koutsoyiannis and Montanari (2007) have shown that the statistical uncertainty of the mean value of (hydrological) time series is increased in the presence of correlations, and in particular long-range correlations.



Long-range correlation is also reflected by a scaling of the power spectral density, $S$ (square of the modulus of the Fourier coefficients appropriately normalized) with frequency $f$. The power-spectral density $S$ exhibits a power-law scaling such that (Taqqu and Samorodnitsky, 1992; Beran, 1994):

$$S(f) \sim f^{-\beta}, \tag{10}$$

where $f$ is the frequency and the relationship holds (for long-range persistence) over all $\beta$ (Pelletier and Turcotte, 1999; Witt and Malamud, 2013). Positive exponents ($\beta > 0.0$) in **Eq. (7)** represent positive (long-range) persistence and negative ones ($\beta < 0.0$) anti-persistence. The specific case of $\beta = 0.0$ corresponds to an uncorrelated time series (e.g., a white noise), and a value of $\beta = 1.0$ is known also as a 1/f or pink (Mandelbrot and van Ness, 1968; Bak et al., 1987). In this paper, to avoid confusion with other estimators (e.g., DFA, see next), we will indicate the measurement of the strength of long-range persistence by power-spectral analysis using the notation $\beta_{PS}$.

Another common method for quantifying long-range correlations is Detrended Fluctuation Analysis (DFA) (e.g., Peng et al., 1994; Kantelhardt et al., 2001). Here, the scaling properties of long-range correlated time series are quantified in terms of the fluctuation function. In this paper the number of floods per year, decade or century can be analysed by DFA, we call these time series here $x_t$, $t = 1, \ldots, N$. The fluctuation function is based on the running sums (or profile) of the considered time series $x_t$, $t = 1, \ldots, N$:

$$s_t = \sum_{i=1}^{t} x_i . \tag{11}$$

This time series of the running sums $s_t$, $t = 1, \ldots, N$ is then split up into non-overlapping segments of length $l$. For the $k^{th}$ segment of the running sums, $s_{k,i}$ , $i = 1, \ldots, l$, the fluctuation is determined as the variance of the difference of this segment and its best fitting polynomial trend $t_{k,i}$ , $i = 1, \ldots, l$, (with the polynomial order $k$, usually between 1 and 4)

$$F^2(k,l) = \frac{1}{l} \sum_{i=1}^{l} \left( s_{k,i} - t_{k,i} \right)^2 . \tag{12}$$

The fluctuation of the time series is the mean of the fluctuation of the segments:

$$F^2(l) = \left\langle F^2(k,l) \right\rangle_k . \tag{13}$$

Where $\left\langle \ \right\rangle_k$ is the mean value taken over all fluctuations of length $k$ segments. If the underlining time series $x_t$, $t = 1, \ldots, N$ has long-range correlations, then the fluctuation function $F(l)$ exposes for long segment lengths $l$ a power-law scaling and will scale as (Peng et al., 1992).

$$F(l) \sim l^{\alpha} . \tag{14}$$



The strength of long-range correlations, $\beta$, is related to the scaling parameter of the fluctuation function, $\alpha$, as $\beta = 2\alpha - 1$. If polynomials of order $k$ are considered, then the resultant estimate of the long-range dependence is called DFA$k$ (e.g., DFA1, DFA2). In this paper we will calculate $\alpha$ using DFA1 to DFA4. We will provide results using the notation $\beta_{\mathrm{DFA}}$ (to indicate the use of DFA) based on $\beta_{\mathrm{DFA}} = 2\alpha - 1$.

Extensive details about power spectral analysis, DFA and software written in R (R Core Team, 2013) for performing time series analysis using these techniques are given in the review paper by Witt and Malamud (2013). This latter study also investigates biases of both techniques which typically occur when they are applied to time series that have a one-point probability distribution that is strongly non-Gaussian and or the time series has very few values (e.g., just a few thousand data).

The idea of long-range correlations is commonly used to characterize observational and experimental measurement data across many scientific disciplines: For instance Peng et al. (1993b, 1994) found long-range persistence in the nucleotide organisation of DNA sequences, Kurths et al. (1995) detected long-range correlations in time series from radio astronomy, Peng et al. (1993a) and Penzel et al. (2003) analysed long-term recordings of heart rate variability, Koscielny−Bunde et al. (2006) found long-range persistence in river runoff series, Fraedrich and Blender (2003) confronted strength of long-range correlations of weather records and weather simulations and Ghil et al. (2011) presented an example of a long-range persistence analysis of an unequally spaced time series (the Nile River). Many more examples are given in Witt and Malamud (2013). Long-range correlation research is not restricted to time series with continuous one-point probability distribution. These correlations have also been investigated in integer-valued time series. For example, the example given above for Peng et al. (1993b, 1994) research on long-range persistence of DNA's nucleotide organisation represented the DNA as sequences with four different types of symbols. In another two examples, Altmann et al. (2012) studied correlations in texts as represented by binary sequences of texts and Schaigorodsky et al. (2014) investigate long-range memory in the opening moves of chess games.

Fractional noises are standard examples of long-range correlated time series (Malamud and Turcotte, 1999; Mandelbrot, 1999), which we will use here to help us model long-range persistent characteristics of our palaeoflood time series. Examples of six fractional noises are shown in **Fig. 6**, with the strength of long-range persistence ranging from $\beta = 0.0$ (a white noise) to $\beta = 1.0$ (a pink noise), in 0.2 increments. In the white noise in **Fig. 6**, the values are uncorrelated with each other, and as we increase the strength of long-range persistence, the values, although drawn from the same underlying probability distribution, become more correlated with one another (and their clustering increases).

### 3.4.2 Application of DFA and power-spectral analysis to our palaeoflood time series

We now apply both DFA and power-spectral analysis to our time series given in **Fig. 3.** We start with our yearly flood record $n_{\mathrm{year}}$ which contains 771 floods and extends over 9336 yr. It has a heavily skewed one-point probability distribution with a





coefficient of variation of $c_v = 3.46$. Witt and Malamud (2013) showed that very asymmetric one-point probability distributions lead to a systematic underestimation of the persistence strength for both techniques. Due to the palaeoflood time series elements being integer valued and in a small range ($0 \leq n_{year} \leq 4$ floods yr$^{-1}$) both techniques lead to uncertain results. Therefore, we will not apply power spectral analysis and DFA to the annual palaeoflood data set. But we will quantify long-range persistence

of the decadal and centennial flood series, as the data are less heavily skewed ($c_v = 1.30$ and $0.77$, respectively) and where there are a wider range of integer values ($0 \leq n_{decade} \leq 8$ floods decade$^{-1}$ and $0 \leq n_{century} \leq 31$ floods century$^{-1}$).

In **Fig. 7** we show the results of the DFA analysis of $n_{decade}$. The fluctuation functions for different orders of detrending are shown on logarithmic axes. The almost linear shape indicates power-law behaviour of the fluctuation function for DFA3 and

DFA4. DFA1 and DFA2 follow this linear shape just for segment lengths up to 70 or 100 decades. We expect a slowly varying trend (a long-term cyclicity >700−1000 yr) to cause this dependence on the type of detrending and will analyse it in more detail in the next **Sect. 3.5**. The power-law exponent was estimated (for DFA3) as $\alpha = 0.62$ which corresponds to a strength of long-range persistence of $\beta_{DFA} = 0.25$. This is a weak strength of long-range persistence. The third data set that counts the number of floods per century contains just 92 time series elements, which is not enough for a reliable DFA.

Power spectral analysis for the decadal palaeoflood series $n_{decade}$ is shown in **Fig. 8**. Although the power spectral density $S$ has a lot of scatter, it is clearly oriented along a line in the log-log axes, and thus power-law (self-affine) behaviour. The persistence strength (i.e., the magnitude of the power-law exponent of the power spectral density $S$) is estimated as $\beta_{PS} = 0.39$. This again indicates a weak strength of long-range persistence. The centennial palaeoflood series $n_{century}$ is not suited for power spectral

analysis due to the short length of $N_{century} = 92$ centuries.

We also see the hint of long-term periodic components superimposed on the long-range persistence that is identified using power spectral analysis. In **Fig. 8** are given three 95% confidence intervals. The derivation of these confidence intervals will be described in detail in a later section (**Sect. 4.2**). What we observe is that three values (large blue dots, outlined with a grey

ellipse) are above the three 95% confidence intervals (i.e., they are significant with $p < 0.05$) at a cyclicity of 1300 to 2300 yr. Again, as for DFA, we will explore this long-term cyclicity (fluctuation) that appears to be superimposed on the long-range persistent signal in the next subsection.

Both DFA and power-spectral analysis, however, have systematic and random errors in the resultant estimator for time series

with very asymmetric probability distributions and the finite size effects of the time series. The systematic error is how much the resultant estimator deviates from the underlying 'correct' value, and random errors are the spread of the estimated quantity. Witt and Malamud (2013) studied systematically these two types of errors for these two techniques. Using the techniques given in Witt and Malamud (2013) and ignoring any underlying periodicity as a contributing factor to any errors, for our palaeoflood





per decade time series with 927 values and a coefficient of variation of $c_v = 1.3$, $\beta_{PS}$ would lead to a calibrated $\beta_{PS}^* = 0.52$ with

(0.32, 0.71) the 95% confidence interval and calibrating the value of $\beta_{DFA}$ would give $\beta_{DFA}^* = 0.37$ with (0.13, 0.63) the 95%

confidence interval.

However, we have already indicated that there might be a long-range fluctuation (cyclicity) superimposed on our long-range persistence signal. Therefore, we are unable to 'calibrate' our $\beta_{PS}$ and $\beta_{DFA}$ using the techniques of Witt and Malamud (2013) and will instead proceed with an investigation of the period and amplitude of the long-term cyclicity (see **Sect. 3.5**) which will lead us to a model-based approach (see **Sect. 4**). In summary, by applying DFA and power spectral analysis we find evidence for long-range correlations in our palaeoflood data set. The estimated strength of persistence for the (uncalibrated) values of $\beta_{PS} = 0.39$ and $\beta_{DFA} = 0.25$.

### 3.5 Cyclicity and fluctuations in the 9.3 kyr palaeoflood record

In **Sect. 3.4** we quantified the long-range persistence of our three palaeoflood records. We also found that by applying DFA [power-spectral analysis] there was an indication [hint] for a long-term cyclicity at $>700-1000$ yr [1300−2300 yr] time scales. We now further explore potential long-term temporal cycles and fluctuations in our data.

Several climate cycles on millennial scales are known from the analysis of long-term palaeo climate proxy data (e.g. ice cores). They are related to Heinrich events (Heinrich, 1988; Bond et al., 1992) and the Dansgaard-Oeschger (Dansgaard et al., 1993, Johnsen et al., 1992; Voelker, 2002) cycle which are caused by ocean ice dynamics and variations in solar activity (Gray et al., 2010). The corresponding cycle lengths range between 1000−5000 yr. These long-term cyclicity processes *might* play a role as a background signal in the flood frequency described in our palaeoflood data set.

Previously (**Sect. 3.2**), our modelling considered the flood rate per year as constant ($\lambda_{year} = 0.83$ floods yr$^{-1}$) (see **Eq. (1)**, **Fig. 3**, **Fig. 5**). We will now model the flood rate $\lambda_{year}$ as a function of time $t$ (in yr), and use the symbol $\Lambda(t)$, $t = 1, \ldots, 9336$ yr, to denote the long-term fluctuations of the flood rate in our palaeoflood record. First, to visual the long-term fluctuations of the flood rate, we use a kernel density estimator (see **Appendix B**) which computes a weighted mean of the number of floods per year $n_{year}$ for a sliding time window. We applied Gaussian (based on a standard deviation of $\sigma = 250$ yr) weights and dealt appropriately with the gap and the values close to the boundaries of the observational interval and also computed 95% confidence intervals for $\Lambda$ (for technical details see **Appendix B**). The results (**Fig. 9A**) show that the time-dependent flood rate $\Lambda$ is varying on millennial scales. We have high values of $\Lambda > 0.15$ floods yr$^{-1}$ at the beginning of the record, a first minimum with $\Lambda = 0.065$ floods yr$^{-1}$ at a relative age of $t = 1,000$ yr, a second maximum at $t = 2,000$ yr, a second minimum at $t = 3,400$ yr and an absolute maximum with $\Lambda = 0.16$ floods yr$^{-1}$ at a relative age of $t = 4,300$ yr. The long-term fluctuations





of the flood rate $\Lambda(t)$ continue for larger relative ages, but get weaker. As the 95% confidence intervals of the absolute maximum and some of the minima do not overlap, the fluctuations are statistically significant. This long-term fluctuation can also be seen in the time series of $n_{century}$ (**Fig. 9B**), but due to scatter the millennial-scale cyclicity is not well defined.

We then model the cyclical behaviour of the flood frequency by fitting a sinusoidal function to the annual palaeoflood time series $n_{year}$. A best-fit sinusoidal function is obtained by applying (with period $T$) least-squares regression to time $t$ in yr:

$$n_{year}(t) = a_1 + a_2 \sin(2\pi t/T) + a_3 \cos(2\pi t/T) + \varepsilon(t) \tag{14}$$

with $a_1$ a constant very similar to the mean flood rate $\lambda_{year} = 0.083$ floods yr$^{-1}$ given in **Eq. (1)**, the term "$a_2 \sin(2\pi t/T) + a_3\cos(2\pi t/T)$" is the best-fit sinusoidal function with optimized parameters $a_2$ and $a_3$, and $\varepsilon(t)$ is the residual of the statistic

model. Using trigonometric functions, we can rewrite the middle two terms of the right hand side of **Eq. (14)** such that:

$$n_{year}(t) = a_1 + A_{rate}\sin((2\pi t / T) + \phi) + \varepsilon(t). \tag{15}$$

with the amplitude $A_{rate} = (a_2{}^2 + a_3{}^2)^{0.5}$ and the phase $\phi = \arctan(a_3/a_2)$.

We have calculated the best-fitting sinusoidal model for periods $1500 < T < 2500$ yr (with step size $\Delta T = 10$ yr) and have found

that the model leads to the smallest variance of the residuals ($\sigma^2(\varepsilon^2)=0.081$) when $T = 2030$ yr (see **Appendix C Fig. C1**), where we find just one minimum and a very low curvature (platykurtic) shape. The model with $T = 2030$ yr best explains the original palaeoflood time series when considering the time dependent flood rate as a sine wave. The corresponding model parameters are (when $T = 2030$ yr) $a_1 = 0.082$ floods yr$^{-1}$ ($\approx \lambda_{year} = 0.083$ floods yr$^{-1}$), amplitude $A_{rate} = 0.049$ floods yr$^{-1}$, and phase $\phi = 39°$. The value for $A_{rate}$ is almost 60% of the constant palaeoflood rate ($\lambda_{year}$). This model is graphically presented in

**Fig. 9** as a red line.

When the periodic rate model is compared to the kernel density estimate of the time dependent rate $\Lambda$, we find (i) a good agreement of the positions of the maxima and minima, (ii) constant values of the maxima and of the minima of the periodic rate model but very different values for the five maxima of the time dependent flood rate $\Lambda$ and (iii) the periodic rate model is

far outside the 95% confidence levels of the time dependent rate $\Lambda$ for the time interval of $5800$ yr $< t < 6700$ yr, which is just 10% of the length of the palaeoflood record. We can conclude that the periodic rate model captures the most important features of the time dependent rate $\Lambda$.

### 3.6 Summary of clustering, correlation and cyclicity of the 9.3 kyr Piànico-Sèllere palaeoflood record

In **Sect. 3** we have seen that the decadal number of floods cannot be explained by a Poisson process due to the palaeoflood

time series' one-point probability distribution and long-range persistence (strength $\beta_{PS} \approx 0.39$ using power spectral analysis and $\beta_{DFA} \approx 0.25$ using DFA). Furthermore, we found our palaeoflood series to be clustered as the interevent occurrence times





are approximately Weibull distributed (shape parameter $k_W = 0.78 \pm 0.02$) and long-range correlated (power-law exponent of the autocorrelation function $\gamma_\Delta = 0.45 \pm 0.14$). We also found evidence for long-term cyclicity (period $T \approx 2030$ yr). In the next section we will suggest a minimal model that captures these identified characteristics of clustering, correlation and cyclicity.

## 4 Creating a peaks over threshold model to capture correlations and clustering of the observational data

In the previous section, we examined a 9.3 kyr palaeoflood record from Piànico-Sèllere that occurred sometime during the period 400 to 700 ka, and found long-range persistence indicated by power-law behaviour of the power spectral density, the fluctuation function and the autocorrelation function as well as temporal clustering and long-period cyclicity of the flood time series. We now introduce a model using peaks over threshold (POT) of synthetic time series that consist of a fractional Gaussian noise (FGN) + period, which we will abbreviate as a $POT_{FGN+Period}$ model. This model is based on the idea of long-range persistence and cyclicity that (i) captures the correlation and clustering properties of the palaeoflood observational data, (ii) captures the long-period cyclicity, (iii) depends on a minimum number of parameters. In this section we will first present a method by which we can create a general model to incorporate correlations, clustering and cyclicity (**Sect. 4.1**). We next quantify the correlation, clustering and cyclicity properties of this model (**Sect. 4.2**) and specify the parameters of this general model to reflect the specific properties of our observed 9.3 kyr palaeoflood data (**Sect. 4.3**). We then confront the model with the optimized parameters with the palaeoflood data (**Sect. 4.4**) and finally discuss properties and a use of this specified model by analysing long model realisations (**Sect. 4.5**).

### 4.1 A peaks over threshold model utilizing fractional noises and long-term cyclical behaviour ($POT_{FGN+Period}$)

For describing our palaeoflood record, we model a series of events with no magnitude along a time line such that they exhibit correlation and clustering. As discussed in **Sect. 3.5**, fractional noises exhibit linear correlations (i.e., by linear, we mean correlations that can be quantified by autocorrelation function or power spectral analysis) that are long-range persistent (Beran, 1994). Peaks over thresholds (POT) of fractional noises are a popular technique for modelling time series of extreme events, which expose long-range persistence. To model this type of linear correlations in time, extreme events are considered to be the values of the considered fractional noise that exceed a high threshold (*i.e.*, the POTs) (Altmann and Kantz, 2005; Bunde et al., 2005; Eichner et al., 2007; Olla, 2007; Santhanam and Kantz, 2008; Moloney and Davidsen, 2009). It has been shown, that the interevent occurrence times of such POT values follow a heavy-tailed or Weibull distribution and have long-term correlations (Bunde et al., 2005). Similar results have been found to hold for processes with multifractal and other nonlinear correlations (Bogachev et al., 2007, 2008; Olla, 2007), but not for nonlinear deterministic systems (Schweigler and Davidsen, 2011). We have taken the idea of using POTs of fractional noises and have modified it in order to gain long-term periodic fluctuations in the flood frequency and an annual number of floods ranging from $0 \leq x_{year}(t) \leq 5$ floods yr$^{-1}$.





The long-period cyclical behaviour of our 9.3 kyr palaeoflood data set was taken into account by superimposing a sine wave with a period of $T = 2030$ yr on an input fractional Gaussian noise. The resulting time series is formally different from the periodic fractional noises introduced by Montanari et al. (1999). An integer valued time series of floods per year was created from the strongly fluctuating input signal by introducing a group of thresholds. The model POT$_{FGN+Period}$ was implemented

using the following three-part schema:

(i) *Creation of a superposition of a fractional noise and a periodic signal (FGN+Period) to input to our Peaks over Thresholds (POTs) model.*

    a. Begin with a realisation $y_t$, $t = 1$ ,…, 9270 yr of a fractional Gaussian noise (FGN) with a given long-range

persistence strength ($\beta_{model}$).

    b. Normalize the FGN so that it has mean = 0 and standard deviation 1.0.

    c. Add to the normalized FGN a sine wave with period $T = 2030$ yr and a given amplitude (which can be varied) $A_{model}$. The $T = 2030$ yr is based on the results of **Sect. 3.6**. The result of the normalized FGN superimposed with a sine wave is a periodic FGN.

*(ii) POTs to obtain the number of floods per year.*

    a. Begin with our periodic FGN from Step 1.

    b. Decide the maximum number of floods per year, $x_{max}$, the model will produce. For this paper, we used $x_{max}$ =5 floods yr$^{-1}$, *i.e.*, the model can produce $x_{year} = 0, 1, 2, 3, 4, 5$ floods yr$^{-1}$.

    c. Define $x_{max}$ thresholds where the thresholds $\theta_{01}$, $\theta_{12}$, $\theta_{23}$, …, are $x_{max}$ horizontal lines which will intersect our

periodic FGN. The thresholds $\theta_{01}$, $\theta_{12}$, $\theta_{23}$ … are chosen such that the resultant one-point probability distribution is a Poisson distribution with a rate parameter of $\lambda_{year} = 0.083$ floods yr$^{-1}$ , i.e., the same as in our palaeoflood series

    d. Determine for each value of the periodic FGN $y_t$, $t = 1$, …, 9270 yr the highest of the $x_{max}$ thresholds which is exceeded. The ordinal number of this threshold is the modelled number of floods per year $x_{year}$ (as

illustrated in **Fig. 10B**). If no threshold is exceeded, the number of floods per year is set to zero, $x_{year}(t) = 0$ floods yr$^{-1}$.

    e. The result is an integer valued time series that models the number of floods per year (as illustrated in **Fig. 10C**) that has the same one-point probability distribution, the same number of data points and the same long-term period ($T = 2030$ yr) as our original palaeoflood record. The model depends on two model parameters

(long-range persistence strength $\beta_{model}$ and amplitude of periodic signal $A_{model}$).

(iii) Number of floods per decade and century.

    a. For computing the modelled number of floods per decade $x_{decade}$ the number of floods per year $x_{year}$ is summed up over 10 consecutive years.





b.  For computing the modelled number of floods per century $x_{century}$ the number of floods per decade $x_{decade}$ is summed up over 10 consecutive centuries.

The resultant outputs of our POT$_{FGN+Period}$ model, the integer valued synthetic flood time series $x_{year}$ depend on two parameters

which are both related to the strongly fluctuating input signal: the strength of long-range persistence $\beta_{model}$ and the amplitude of the superimposed sine wave $A_{model}$. In **Fig. 11** twelve model realisations are shown. The six floods per year time series presented in **Fig. 11A** are constructed with strength of long-range persistence $\beta_{model} = 0.0, 0.2, \ldots, 1.0$, but without a long-term periodic component ($A_{model} = 0.0$). We see visually from **Fig. 11A** that higher values of persistence lead to stronger clustering of the annual number of floods, *i.e.*, years with many floods tend to follow years with many floods and years without floods

tend to follow years without floods. The six synthetic floods per year time series presented **Fig. 11B** are constructed with white noise ($\beta_{model} = 0$) and with an increasing amplitude of the long-term periodic component $A_{model} = 0.0, 0.2, \ldots, 1.0$. In **Fig. 11B** we can visually observe that for high values of $A_{model}$ the years with many floods cluster periodically with a period of $T = 2030$ yr.

We now return to the confidence intervals in **Fig. 8**. These were derived as follows: We simulated 1000 realizations of the model POT$_{FGN+Period}$ with $A_{model}=0.0$ and $\beta_{model} = 0.1, 0.3$ or $0.5$. For a fixed value of $\beta_{model}$ the power spectral density (the periodogram) was computed for all resultant event time series. For each frequency $f$ the 1000 values of $S(f)$ were ordered from smallest to biggest and the $975^{th}$ of 1000 values was taken as the upper end of the 95% confidence interval and drawn as a coloured dot in **Fig. 8**.

In summary, the time series presented in **Figs. 11A** and **B** show that high values of either of the model parameters lead to strong clustering. In the next subsection, we will investigate the dependence of the clustering and correlation properties on the model parameters.

## 4.2 Quantifying correlation, clustering and cyclicity properties of the POT$_{FGN+Period}$ model

In the previous subsection, we have shown that it is possible to use the POTs of a fractional noise that has been superimposed with a 2030 yr period to create an integer valued time series with long-range persistence (correlation) properties and periodicity. In addition, we have been able to capture the same 'number' of values in our synthetic flood time series as in our palaeoflood record.

Now we want to evaluate the strength of clustering caused by persistence and the contribution of the long-term periodicity for different model parameters of our POT$_{FGN+Period}$ model. This is done by creating multiple model realisations for different pairs of model parameters and by measuring the strength of long-range persistence, the shape parameter of the Weibull distribution





of the interevent occurrence times and the periodic rate fluctuations of these realisations and comparing them with the values measured for the palaeoflood record. The model has two parameters: (i) $\beta_{\text{model}}$, the *long-range persistence strength* of the underlying fractional Gaussian noise, and (ii) $A_{\text{model}}$, the *amplitude of the superimposed periodic component* with a period $T = 2030$ yr. Both parameters were sampled with a step size of 0.02 from 0.00 to 1.00, resulting in a square grid of $51 \times 51 = 2601$ pairs of model parameters. For each pair of model parameters ($\beta_{\text{model}}$, $A_{\text{model}}$) 1000 synthetic flood series (model realisations) were produced using the peaks-over-thresholds method described in **Sect. 4.1** and illustrated in **Fig. 10**. The differences between the realisations for one and the same pair of model parameters are caused by different noise realisations.

The amplitude of the best-fitting sinusoidal flood rate $A_{\text{rate}}$ assuming a period of $T = 2030$ yr and the shape parameter $k_{\text{W}}$ of the best-fitting Weibull distribution of the interevent occurrence times were determined for each model realisation ($x_{\text{year}}$). Further, each resultant palaeoflood yearly series ($x_{\text{year}}$) was aggregated to give the number of floods per decade ($x_{\text{decade}}$) and then (using power-spectral analysis and detrended fluctuation analysis) its strength of long-range persistence $\beta_{\text{PS}}$ and $\beta_{\text{DFA}}$ was computed.

In **Fig. 12A** to **12D**, for each of the $51 \times 51$ 'cells' of the parameter space of our POT$_{\text{FGN+Period}}$ model, are given the mean values of measured $\beta_{\text{PS}}$, $\beta_{\text{DFA}}$, $k_{\text{W}}$ and $A_{\text{rate}}$ (see legend for colours and contours) for that cell's ($\beta_{\text{model}}$, $A_{\text{model}}$) 1000 model realisations. The graphs show that on average an increase of the model parameter *long-range persistence strength* of the underlining fractional Gaussian noise, $\beta_{\text{model}}$, leads to an increased strength of long-range persistence, $\beta_{\text{PS}}$ and $\beta_{\text{DFA}}$, of the synthetic floods per decade series $x_{\text{decade}}$ and to a decrease of the shape parameter $k_{\text{W}}$ of the Weibull distribution of the modelled interevent occurrence times (of $x_{\text{year}}$). Moreover, an increase of the $A_{\text{model}}$ parameter of our POT$_{\text{FGN+Period}}$ model (*i.e., the amplitude of the superimposed periodic component* with period $T = 2030$ yr), leads to a larger best fitting sinusoidal flood rate $A_{\text{rate}}$. However, in both cases the intensity of the increase depends also on the second model parameter, as otherwise, the images presented in **Fig. 12A** to **12D** would have horizontally or vertically striped structures. Furthermore, the mean values of $\beta_{\text{PS}}$, $\beta_{\text{DFA}}$, $k_{\text{W}}$ and $A_{\text{rate}}$ (**Fig. 12A** to **12D**) do not give information about the spread of these measured parameters for a specific model parameter. This implies that for a specific cell, we do not have the information about the probability to have a model realisation that is similar to the Piànico palaeoflood data set.

### 4.3 Parameter fitting of the POT$_{\text{FGN+Period}}$ model

Here we identify pairs of model parameters ($\beta_{\text{model}}$, $A_{\text{model}}$) that correspond to flood time series whose strength of long-range persistence ($\beta_{\text{PS}}$ or $\beta_{\text{DFA}}$), interevent occurrence time distribution (measured as the shape parameter of the Weibull distribution $k_{\text{W}}$) and best fitting sinusoidal flood rate $A_{\text{rate}}$ have values close to those measured for the palaeoflood record from Piànico. From the $2.6 \times 10^6$ synthetic flood series realisations generated in **Sect. 4.2**, the realisations' parameters are within the range of our palaeoflood original series parameters as follows: 162,213 realisations (6.2% of the total # of model realisations) with $0.370 \leq \beta_{\text{PS}} \leq 0.410$, 157,782 realisations (6.1%) with $0.23 \leq \beta_{\text{DFA}} \leq 0.27$, 132,025 realisations (5.1%) with $0.76 \leq k_{\text{W}} \leq 0.81$



and 152,056 realisations (5.8%) with $0.0450 \leq A_{\text{rate}} \leq 0.0530$. In **Fig. 12C** the corresponding model parameters of these realisations are presented as clouds of points and each point stands for a $POT_{\text{FGN+Period}}$ model realisation that is similar to the Piànico palaeoflood record with regard to a single measure of correlation or clustering. These points are located in specific areas: For instance, model realisations with a strength of long-range persistence $\beta_{\text{PS}}$ similar to the value of the palaeoflood record (pink dots) are found for $\beta_{\text{model}} < 0.8$ and on the right hand side of the line connecting the points ($\beta_{\text{model}} = 0.4$, $A_{\text{model}} = 0$) and ($\beta_{\text{model}} = 0.0$, $A_{\text{model}} = 0.8$). Realisations with values of $A_{\text{rate}}$ close to 0.049, *i.e.*, the value of the best fitting sinusoidal flood rate of the palaeoflood record, are centred around $A_{\text{model}} = 0.32$ (green dots) with a spread that is growing with increasing values of $\beta_{\text{model}}$. Next, we will identify the model parameters $\beta_{\text{model}}$ and $A_{\text{model}}$ which lead to model time series that best match our palaeoflood record.

**Figure 12E** also shows that there are just a very few model parameters that belong to all four clouds and thus are in the desired ranges of $\beta_{\text{PS}}$, $\beta_{\text{DFA}}$, $k_{\text{W}}$, $A_{\text{rate}}$. To identify optimum model parameters, we choose $POT_{\text{FGN+Period}}$ model realisations where the following four parameters intersect: $\beta_{\text{PS}}$ and $\beta_{\text{DFA}}$ (characterise strength of long-range persistence, pink and brown dots, respectively), $k_{\text{W}}$ (characterizes clustering, blue dots) and $A_{\text{rate}}$ (characterises cyclicity strength, green dots). In other words, we find where the brown, pink, blue and green dots intersect. Just 143 realisations (presented as black dots in **Fig. 12E**) out of the the $2.6 \times 10^6$ model realisations are part of this intersection. They are located in a small area of the 2D parameter space: $0.10 \leq \beta_{\text{model}} \leq 0.30$ and $0.20 \leq A_{\text{model}} \leq 0.38$. It should be noted that this area does not intersect with the *x*- or the *y*-axis, meaning that both model parameters are necessary for an appropriate description of the palaeoflood data. Because 143 realisations were not sufficient to effectively investigate (and visualize) the 2D probability distribution of $\beta_{\text{model}}$ and $A_{\text{model}}$, for $0.06 \leq \beta_{\text{model}} \leq 0.34$ and $0.16 \leq A_{\text{model}} \leq 0.42$ (a slightly extended area as to that just found) we created more realisations (10,000 per cell) to gain a total of 1518 realisations with the desired long-range-persistence, clustering and cyclicity properties. The 2D probability density of these points has been approximated by a normalized 2D histogram (**Fig. 12F**). The grid cell of $\beta_{\text{model}} = 0.25$ and $A_{\text{model}} = 0.30$ (presented as a red bullet) has the highest probability. This pair of model parameters is considered to the best fitting. Approximately 50% of the 1518 model realisations that are similar to the palaeoflood data set are found for model parameters $0.19 < \beta_{\text{model}} < 0.29$ and $0.26 < A_{\text{model}} < 0.34$; this range is indicated by blue dashed lines in **Fig. 12E**. These ranges give an estimate of the accuracy of $POT_{\text{FGN+Period}}$ model parameter determination.

In summary, the model evaluation provides evidence that both model parameters of the strongly fluctuating input signal (the strength of long-range persistence of the fractional noise and the amplitude of the sine wave superposed on it) of our $POT_{\text{FGN+Period}}$ model are required for an appropriate modelling of the clustering properties of the palaeoflood record. Optimum parameters including error bars ($\beta_{\text{model}} = 0.24 \pm 0.05$ and $A_{\text{model}} = 0.030 \pm 0.04$) have been determined. One realisation of the model with optimum parameters is given in **Fig. 13A**.



### 4.4 Model confrontation

In the last subsection, we specified the parameters of our $POT_{FGN+Period}$ model in order to reproduce the observed clustering and long-term periodic properties. Now we will investigate further model properties such as the distributions of floods per decade or per century and confront these properties with the values measured for the palaeoflood record from Piànico.

For a comparison of the properties of the palaeoflood data and of the $POT_{FGN+Period}$ model with the specified parameters, we have created 1000 model realisations (50 of these $x_{year}$ series shown in **Fig. 13B**). For each of the 1000 model realisations the time dependent flood rate $\Lambda_{model}(t)$ (**Sect. 3.6**) was computed which enabled the calculation of the mean time dependent flood rate $\overline{\Lambda}_{model}(t)$ and its 95% confidence intervals (**Fig. 13C**). The mean time dependent flood rate $\overline{\Lambda}_{model}(t)$ fluctuates

periodically with a 2030 yr period and replicates the long-term changes in rate fluctuations qualitatively. However, the amplitudes of the rate changes for some time intervals ($t = 0$ to 600 yr, 3800 to 4800 yr and 5500 to 6800 yr) are not correctly captured, i.e., they are outside the 95% confidence intervals of $\Lambda_{model}$. This is due to the stationary character of our palaeoflood model and the non-stationary character of the long-term-rate fluctuations.

For the ensemble of all model realisations, the histograms of the number of floods per year, per decade and per century were computed, such that the mean number of floods per time unit with 95% error bars could be estimated (**Fig. 13D**). The number of the Piànico palaeofloods per year, decade and century (as discussed in **Sect. 3.1** and **Fig. 3A**) fit well within the 95% confidence intervals of the considered histograms.

Finally, for the ensemble of all $POT_{FGN+Period}$ (with specified parameters) model realisations, the clustering and correlation properties were quantified: the persistence strength ($\beta_{PS}$, $\beta_{DFA}$), the shape parameter of the Weibull distribution of the interevent occurrence times ($k_W$) and the time dependent flood rate ($\Lambda_{model}(t)$) were computed (**see Table 2**). **Table 2** shows, that the correlation properties ($\beta_{PS}$, $\beta_{DFA}$) of the model are on average weaker than those of the palaeoflood record. Nevertheless, the 95% confidence intervals from the model contain the original palaeoflood values for $\beta_{PS}$, $\beta_{DFA}$, $k_W$, $A_{rate}$. The clustering





properties ($k_W$, $A_{rate}$) of the data are excellently captured by the model as the mean values of the corresponding measures computed for the model realisations are very close to the values measured for the data.

In summary, we find a good agreement between the POT$_{FGN+Period}$ model and the palaeoflood data series in particular with respect to the distribution of floods per time unit and the clustering properties.

## 4.5 Using and analysing realisations from the POT$_{FGN+Period}$ model

We have shown in **Sect. 4.4** above, that our POT$_{FGN+Period}$ model reproduces several properties of the palaeoflood time series including the distribution of floods per year, decade and century as well as long-range persistence, clustering and cyclicity properties. We now use realisations from the POT$_{FGN+Period}$ model with optimized parameters to explore how the number of floods per time unit (e.g. per decade or century) is related to the number of floods in the following time unit (e.g., does a century (decade) with a few floods tend to follow a century (decade) with a few floods). To do this we create one long model realisation with $10^9$ yr ($10^8$ decades, $10^7$ centuries), we computed the probability of floods per century for given numbers of floods in the preceding century.

**Figure 14A** shows the 2D probability distribution of the number of floods per century ($n_{century}(j)$) and the number of floods in the succeeding century ($n_{century}(j+1)$). This distribution is concentrated along the diagonal line $n_{century}(j+1) = n_{century}(j)$ and thus indicates positive correlations in the number of floods per century time series realisations. **Figure 14B** presents the distribution of the number floods per century if the preceding century contained 0, 8 or 16 floods century$^{-1}$. These distributions are unimodal with a systematic shift in the mode and a dispersion that is increasing with the numbers of floods in the preceding century. In other words, the uncertainty of the forecasted number of floods per century increases with the number of floods in the preceding century.

Our results shown in **Fig. 14** imply that low numbers of floods century$^{-1}$ follow law ones, and high ones follow high. In other words, a century with no floods is much more likely to follow a century with no or a very few floods rather than a century with many floods, and a century with many (e.g., 20) floods is much more likely to follow a century with many (e.g., 20) floods rather than a century with no or a very few floods. However, this holds only in the statistical sense and successions of centuries with many floods can certainly follow centuries with very few floods (and vice-versa). For example, in the palaeoflood data set we have found a "jump" from 4 to 24 floods century$^{-1}$ (between the 38$^{th}$ and 39$^{th}$ century). In our model the probability that a century with 24 floods is following a century with 4 floods is 0.02% on average, but will be much higher for time periods with a high amplitude of the periodic component of the input signal. We also observe that in our original palaeoflood time series there is a jump from 25 to 7 floods century$^{-1}$ between the 45$^{th}$ and 46$^{th}$ centuries. A jump of this magnitude in our optimized POT$_{FGN+Period}$ model has a probability of 3.9% based on our long synthetic realisation, and is thus well captured by the model. Furthermore, we found for the POT$_{FGN+Period}$ model, that medium changes in flood frequencies are fairly likely, as





for instance a transition from 25 floods century$^{-1}$ to <10 floods century$^{-1}$ has a probability of >20%. In a similar manner, we have also examined the summary statistics for the number of floods per decade using our long synthetic time series realisation. For the simulated $10^8$ decades, we find between 0 to 13 floods decade$^{-1}$, although >10 floods decade$^{-1}$ is very unlikely. Similar to the centennial data we find indications for correlations that decades with few [many] floods are followed by decades with few [many] floods, i.e., on average, low values follow low ones, and high ones follow high.

## 5 Summary

We have presented a 10 kyr comprehensive flood record at sub-annual resolution, obtained from the varved interglacial Pleistocene sediments of the Piànico-Sèllere Basin. A lacustrine sediment unit of 9.5 m thickness has been considered which consists of an almost continuous succession of about 15,500 varves that are interpreted as annual cycles. Approximately 8% of the varves contain 1−3 detrital layers which are considered to be the result of channelized streamflow that originated in the hills surrounding the Piànico-Sèllere Basin and triggered by extreme precipitation events (Mangili et al., 2005). Our analysis was aimed at understand the temporal succession of detrital layers.

The analysed palaeoflood record is unique as it comprises the relative timings of 771 flood events, is continuous (except a gap of 65 yr, <1% of the length of the 9336 yr time period) and was not affected by anthropogenic influences. This palaeoflood record provides an example of the high natural variability of flood frequency over a long period (e.g., variations of 0−31 floods century$^{-1}$). Because of the comprehensive data set, the temporal succession of palaeofloods could be extensively studied as to its underlying statistics of correlations, clustering and cyclicity. We showed that the correlations of the yearly number of floods over the 10 kyr palaeoflood are long-range with a long-range persistence strength of $\beta_{PS} \approx 0.39$ (power spectral analysis) and $\beta_{DFA} \approx 0.25$ (DFA). These long-range correlations are modulated by a long-term cyclicity with a period of $T = 2030$ yr. The palaeofloods are also shown to be temporarily clustered as the interevent occurrence times are Weibull distributed (with shape parameter of $k_W = 0.78$).

We have derived a model (POT$_{FGN+Period}$) that is based on peaks over thresholds (POT) of a fractional Gaussian noise (FGN) superimposed by a long period signal. This model allows us to construct many realisations of an event series with properties similar to those identified for the original palaeoflood time series. Should more information (e.g., higher resolution, extending the data series in time) come available related to our original palaeoflood series, then the parameters of correlation, clustering and cyclicity are likely to be slightly changed, and a modified model can then be easily created.

We used our model to create 2,600,000 synthetic flood series with different parameters, and then confronted the clustering, correlation and cyclicity properties of our original palaeoflood record with the synthetic series to come up with optimized parameters in our POT$_{FGN+Period}$ model. Based on the optimized POT$_{FGN+Period}$ model we found that it is important when



modelling our original palaeoflood time series, that the model needs to capture both long-range correlations *and* a slowly varying component (periodicity) to capture the correlation, clustering and cyclicity properties of the palaeoflood record. In other words, the observed temporal flood clusters in our 10 kyr time series cannot be explained by either long-range correlations or slow cyclical changes, rather both components need to be present.

As a practical example of the application of our optimized $POT_{FGN+Period}$ model we use it to create long simulations with parameters similar to that of our palaeoflood time series. We show that centuries with no or a few floods tend to follow each other and that centuries with many floods tend to follow centuries with many floods. We also found that the uncertainty of the forecasted number of floods per century increases with the number of floods in the preceding century.

Our research in this paper combines the statistical analysis of correlations, clustering and cyclicity in a very complete and unique interglacial record of floods from the Pleistocene, allowing one to create a model to simulate many realisations with similar parameters to our original series. We believe that this approach is applicable to other environmental event time series (e.g., palaeo-hazards), where event magnitude is unknown, the timings are unequally spaced in time, and where the one-point probability distribution of the number of events per time unit is strongly asymmetric (i.e., non-Gaussian). This is true for many palaeo- and historical environmental event series (e.g., earthquakes, wildfires, volcanic eruptions). We also believe our approach of creating an optimized model that is congruent with the correlation, clustering and cyclicity parameters of the original event series is one that is generally applicable, and useful for creating many 'realisations'.

**Data availability**: We will add a link to the palaeflood data here, which if the manuscript is accepted, will be lodged at the PANGAEA database (https://www.pangaea.de/about/).



## Appendix A: Detrital layer thickness figure

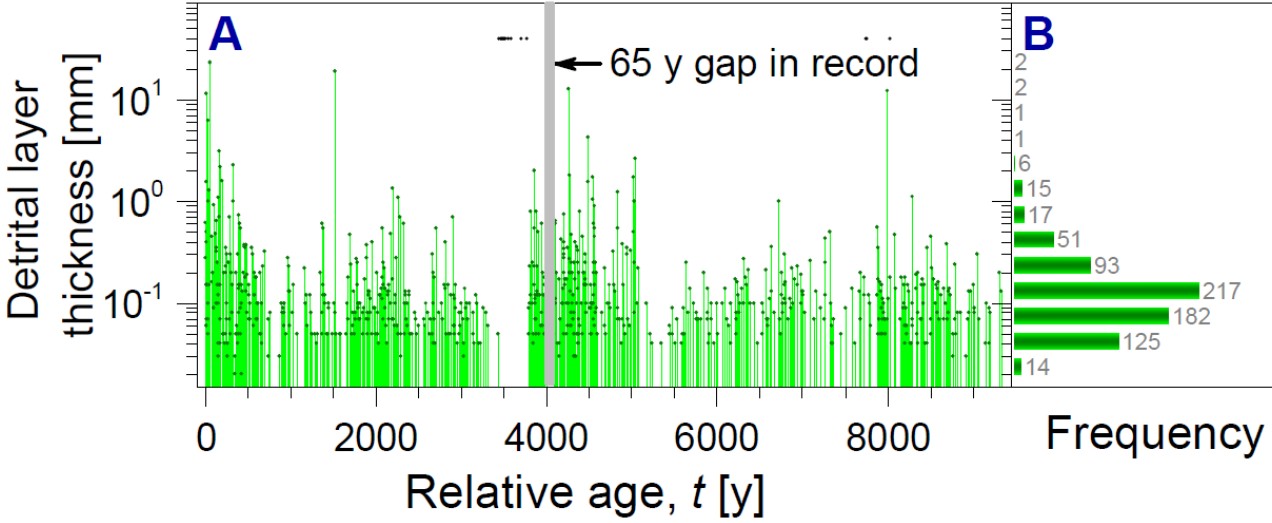

**Figure A1. Detrital layer thickness:** (A) Thickness and temporal location of the detrital layers in the varved sediment. 771 layers are located within the 9271 varves. The x-axis represents relative age, with 0 yr representing the most recent varve and increasing values indicating further back in time. The grey bar represents a sediment gap of 65 yr. The varve thickness (green vertical bars), shown on a logarithmic scale, ranges from 0.002 to 23 mm, some layers are just reported and not measured (black dots). The detrital layers are unequally distributed over time. A histogram of the layer thickness is given in (B)





## Appendix B: Computing the time dependent flood rate with kernel density estimators

Determining the time dependent flood rate is necessary for detecting gradual changes in flood frequency and for identifying long term cyclicity in flood occurrence. Here, the time dependent flood rate and its corresponding 95% confidence intervals have been determined by kernel density estimation (Silverman, 1986). The kernel $K$ is a non-negative function which is

5 symmetric with respect to 0. For our purposes we use a Gaussian kernel $K_G$ with bandwidth σ:

$$K_G(\tau) = \frac{1}{\sqrt{2\pi\sigma^2}} \exp\left(-\frac{\tau^2}{2\sigma^2}\right) \tag{B1}$$

with $\tau$ a discrete time step (for this paper $\tau = 1$yr). The kernel is truncated such that it is defined only for $-3\sigma \leq \tau \leq 3\sigma$. This specific kernel is a discretized and truncated Gaussian probability density function. The bandwidth parameter σ that is the standard deviation of the Gaussian distribution controls the width of the kernel. For this paper, we have applied a bandwidth

of $\sigma = 250$ yr, and therefore the Gaussian kernel that is created has a width of 1500 yr (range of $\pm 3\sigma$). The idea of kernel density estimation is to replace each flood in the floods per year function $n_{year}$ by a smooth kernel and to consider the superposition of all kernels as a measure for the number of floods per year. This is implemented by convolving **Eq. (B1)** with the time series of the number of floods per year, $n_{year}$. At step one, the time dependent flood rate $\Lambda(t)$ with respect to the specific kernel $K_G$ is computed for times $t = 1, \ldots T$ by

$$\Lambda(t) = \left(\sum_{\tau=-3\sigma,\cdots,3\sigma} n_{year}(t-\tau)\mathbf{1}(t-\tau)K_G(\tau)\right) \Bigg/ \left(\sum_{\tau=-3\sigma,\cdots,3\sigma} \mathbf{1}(t-\tau)K_G(t)\right),$$

$$\tag{B2}$$

with **1** the temporal range function of our time series, i.e., $\mathbf{1}(t) = 1$ for values of $t$ in the temporal range of the time series and $\mathbf{1}(t) = 0$ for values outside (i.e., $t \leq 0$, $t > T$ or $t$ is inside in the gap). Thus, the numerator is the convolution of the kernel density $K_G$ and the number of floods per year $n_{year}$ and the denominator is the convolution of the kernel density and the range function of the time series. Note, that the range function is only needed for approximately handling $n_{year}$ values that are near the two

ends of the time series and the gap.

After convolution and derivation of the time dependent flood rate $\Lambda(t)$, in step 2, we now derive their respective 95% confidence intervals. This requires the computation of the effective number of data points $N(t)$ for each time point $t$, which

describes the number of time series elements that contribute to the kernel density estimate of the rate, $\Lambda(t)$:

$$N(t) = 2\sigma \sum_{\tau=-3\sigma,\cdots,3\sigma} \mathbf{1}(t-\tau)K_G(\tau), \tag{B3}$$

This effective bandwidth $N(t)$ is smaller than $2\sigma$ for time points $t$ which are close to the beginning and end of the observational interval (e.g., $t = 1$ or $t = T$), and equal to $2\sigma$ for time points $t$ far from these interval ends.





The distribution of the time dependent rate $\Lambda(t)$ is related to a two parameter Gamma distribution (Johnson et al., 1993). This allows us to calculate the corresponding 95% confidence intervals, i.e., the lower ($\Lambda_{0.025}$) and the upper ($\Lambda_{0.975}$) endpoints of the 95% confidence intervals, as:

$$\Lambda_{0.025}(t) = 2 F_{\Gamma}\left(0.025, 0.5N(t)R(t), 1\right)/N(t)$$
$$\Lambda_{0.975}(t) = 2 F_{\Gamma}\left(0.975, \{0.5N(t)R(t)\}+1, 1\right)/N(t) \text{,}$$

(B4)

5   with $F_{\Gamma}$ the cumulative distribution function of the two parameter Gamma distribution with (i) 0.025 (0.0975) the considered quantile, (ii) 0.5 $N(t)R(t)$ ($\{0.5\ N(t)R(t)\}+1$) the shape parameter and (iii) and 1 the scale parameter. For comparison, we also calculated confidence intervals of the time dependent rates by applying a bootstrap-based method as introduced by Mudelsee (2014) in his section 6.3.2 on inhomogeneous Poisson processes and nonparametric occurrence rate estimations. The two methods for calculating confidence intervals resulted in very similar values.



## Appendix C: Variance of sinusoidal function residuals figure

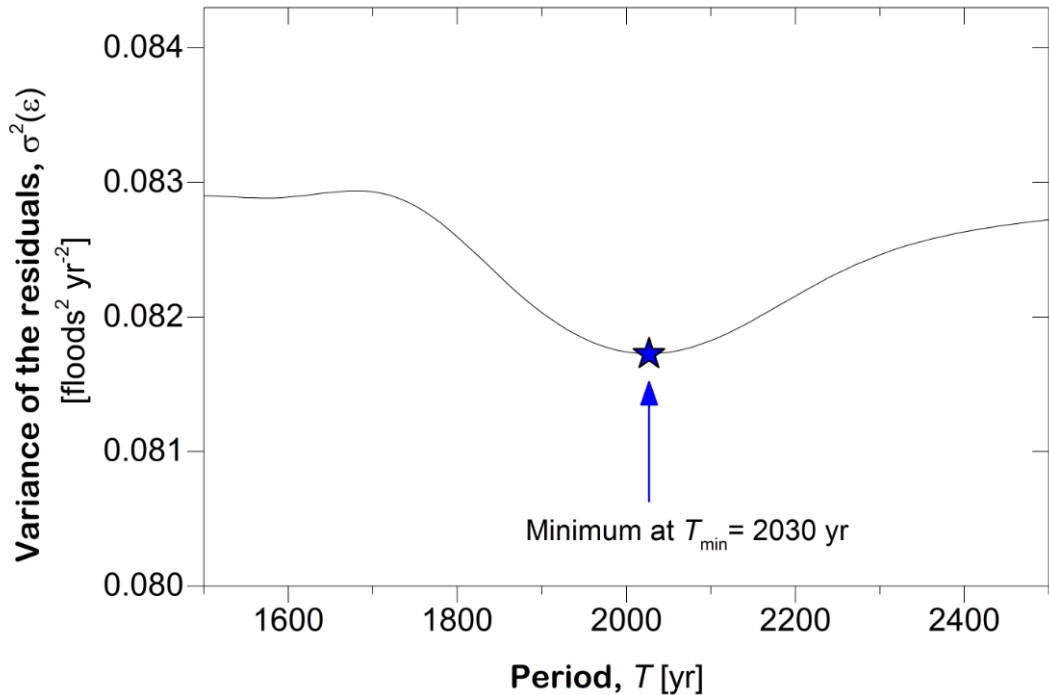

**Figure C1.** Shown is the variance of the residuals ($\sigma^2(\varepsilon)$) for fitting a sinusoidal function to the floods per year series $n_{year}$ as a function of the period $T$ of this periodic function. The function for the variance of the residuals has a minimum at $T_{min} = 2030$ yr (shown as a star in the figure).





**Author contribution**: A. Brauer (AB) and C. Mangili (CM) jointly carried out the sampling and field work for this study. CM carried out varve and detrital layer microfacies analysis under the project supervision of AB. A Witt (AW) and B. Malamud (BDM) did the statistical analyses, model development and theoretical framework, with AW carrying out most of the statistical analyses programming. AW and BDM jointly led the paper for writing and graphical presentation of the results.

**Acknowledgments**: We would like to thank GeoResearch Centre Potsdam (GFZ) G. Arnold, M. Koeler and D. Berger for providing high-quality thin sections.

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





**Table 1: Abbreviations and Notation**

| Abbreviation | Description |
|---|---|
| BVC | *Banco Varvato Carbonatico* (carbonated varved bed) |
| DFA | detrended fluctuation analysis |
| FGN | fractional Gaussian noise |
| MIS | Marine Isotope Stage |
| MLE | Maximum Likelihood Estimation |
| IEOT | interevent occurrence time |
| POT | peaks over threshold |
| PS | power spectrum or power-spectral analysis |

| Symbol | Description | Units |
|---|---|---|
| $\alpha$ | Power-law exponent of the fluctuation function, with $\beta_{\mathrm{DFA}} = 2\alpha - 1$. | [unitless] |
| $\beta_{\mathrm{DFA}}$ | Strength of long-range persistence (based on the power-law exponent $\alpha$ of the fluctuation function) | [unitless] |
| $\beta_{\mathrm{model}}$ | Strength of long-range persistence (model parameter) | [unitless] |
| $\beta_{\mathrm{PS}}$ | Strength of long-range persistence (based on the power-law exponent of the power spectral density) | [unitless] |
| $\gamma$ | Power-law exponent of the autocorrelation function | [unitless] |
| $\Delta, \Delta_j$ | Interevent occurrence time plotted on the x-axis in real time ($\Delta$) and in natural time ($\Delta_j$). | yr |
| $\overline{\Delta}, \overline{\Delta}_j$ | Mean interevent occurrence time $\overline{\Delta} = \overline{\Delta}_j$ | yr |
| $\Theta, \Theta_{12}$ | Threshold (general, that separates 1 and 2 floods per time unit) | [unitless] |
| $\Lambda(t)$ | Time dependent flood rate | [floods yr$^{-1}$] |
| $\Lambda_{\mathrm{model}}(t)$ | Time dependent flood rate of model realizations | [floods yr$^{-1}$] |
| $\overline{\Lambda}_{\mathrm{model}}(t)$ | Mean of time dependent flood rate taken over many model realizations | [floods yr$^{-1}$] |



| Symbol | Description | Units |
|---|---|---|
| $\lambda$ | Rate parameter of the Poisson distribution | [unitless] |
| $\lambda_{year}$, $\lambda_{decade}$, $\lambda_{century}$ | Flood rate per year, per decade, per century | [floods yr$^{-1}$, floods decade$^{-1}$, floods century$^{-1}$] |
| $\lambda_W$ | Scale parameter of the Weibull distribution | [unitless] |
| $\mu$ | Mean of the values considered. | Variable dependent |
| $\sigma$, $\sigma^2$ | Standard deviation, variance | Variable dependent |
| $\tau$ | Time lag in real time | yr, decades, centuries |
| $\tau_\Delta$ | Time lag in natural time (of the interevent occurrence time series $\Delta_j$, $j = 1, 2, …, N$) | [unitless] |
| $A_{model}$ | Amplitude of the periodic component (model parameter) | [unitless] |
| $A_{rate}$ | Amplitude of the periodic component of the time dependent flood rate fit to the palaeoflood time series. | [unitless] |
| $a_1$, $a_2$, $a_3$ | Parameters for the modelling the cyclicity of the annual flood series | [floods yr$^{-1}$] |
| $C(\tau)$, $C(\tau_\Delta)$ | Autocorrelation function, depends on the lags $\tau$ and $\tau_\Delta$ | Variable dependent |
| $c_v$ | Coefficient of variation ($c_v = \sigma/\mu$) | [unitless] |
| $F$ | Fluctuation function | [unitless] |
| $j$ | Index of natural time, $1 \leq j \leq N_\Delta$ | [unitless] |
| $k$ | Integer valued variable | [unitless] |





| Symbol | Description | Units |
|---|---|---|
| $k_{year}$, $k_{decade}$, $k_{century}$ | Number of occurrences of $n_{year}$, $n_{decade}$, $n_{century}$ with a specific value during a given period of time considered. For example, if there are 100 times that $n_{year} \geq 2$ floods $yr^{-1}$ over 10 kyr, then $k_{year} = 100$ yr. | yr decades centuries |
| $k_W$ | Shape parameter of the Weibull distribution | [unitless] |
| $l$ | Integer valued variable | [unitless] |
| $n$ | Intensity (# of occurrences) of the random variable $X$ (in **Eq. (1)**) | [unitless] |
| $n_{year}$, $n_{decade}$, $n_{century}$ | Number of detritic layers (floods) per varve (i.e., per year), per ten varves (i.e., per decade), and per one hundred varves (i.e., per century). | floods $yr^{-1}$, floods $decade^{-1}$, floods $century^{-1}$ |
| $N_{floods}$ | Total number of floods (detritic layers) ($N_{floods} = 771$ floods) | floods |
| $N$, $N_{year}$, $N_{decade}$, $N_{century}$ | Total number of years/decades/centuries where varves are present in the palaeoflood data set ($N_{year} = 9271$ yr, $N_{decade} = 925$ decades, $N_{century} = 92$ centuries) with $N$ the place holder | yr decades centuries |
| $N_{varves}$ | Total number of varves (9336 varves). | varves |
| $N_\Delta$ | Total number of interevent occurrence times, ($N_\Delta = 739$) | [unitless] |
| $P_{exp}$ | One-point probability density distribution of the exponential distribution | [unitless] |
| $P_\lambda$ | One-point probability density distribution of the Poisson distribution | [unitless] |
| $P_W$ | One-point probability density distribution of the Weibull distribution | [unitless] |
| $S(f)$ | Power spectral density depending on the frequency $f$ | Variable dependent |
| $s_t$, $t=1,\ldots,N$ | Running sum | Variable dependent |





| Symbol | Description | Units |
|---|---|---|
| $t = 1, 2, …, N$ | Relative varve index or time index for our palaeoflood time series. Note that the index 'skips over' the 65 yr gap, so for $N = N_{year}$ it goes from $t = 1$ to 9271 yr. | yr |
| $X$ | Discrete random variable in **Eq. (1)** | [unitless] |
| $x_{year}(t)$, $x_{decade}(t)$, $x_{century}(t)$ | Number of floods per year, decade, century depending on time $t$ (model output) | floods yr$^{-1}$, floods decade$^{-1}$, floods century$^{-1}$ |
| $y(t)$ | Model input depending on time $t$ | [unitless] |



**Table 2: Statistics of long-range persistence, clustering and cyclicity properties of the specified ($\beta_{model}$ = 0.24, $A_{model}$ = 0.30) palaeoflood model. 1000 model realisations were created and their long-range correlations ($\beta_{PS}$ and $\beta_{DFA}$), clustering ($k_W$) and cyclicity properties ($A_{rate}$) were quantified. Given is for each measure the mean value and the 2.5th and 97.5th quantile of the model realisations, the value measured for the palaeoflood record and the percentile of the value measured with respect to the values of the model.**

| Measure | Model: mean value (2.5th and 97.5th percentile) | Palaeoflood record | Percentile of palaeoflood record |
|---|---|---|---|
| $\beta_{PS}$ (computed for $n_{decade}$) | 0.26 (0.14, 0.40) | 0.39 | 97% |
| $\beta_{DFA}$ (computed for $n_{decade}$) | 0.21 (0.07, 0.34) | 0.25 | 74% |
| $k_W$ | 0.79 (0.74, 0.84) | 0.79 | 48% |
| $A_{rate}$ | 0.050 (0.038, 0.063) | 0.049 | 48% |





**Figure 1. Example of correlations, clustering and cyclicity (figure after Witt et al., 2010).** (A) Tree ring standardized growth index for Bristlecone pine, White Mountain, California, USA, for the years AD 0–1962 (Ferguson et al., 1994). (B) Cosmic ray neutron counts per hour, Beijing, China, 1 January to 11 March 2008 (NGDC, 2008). In (A) and (B) successive values in each both series are positively correlated with one another, and are examples of persistent time series. (C) The maximum wind speed in knots is plotted as a function of date, for 730 Atlantic Basin named and unnamed severe storms and hurricanes, 1930–2003 (NOAA, 2003); only storms with maximum wind speed ≥ 35 knot are included. (D) The same data as in C, but focussing in on 1980–1990, with minor tick marks representing months. The value of maximum wind speed for each storm event is projected to the x-axis (wind speed = 0 knot). Below this is shown (green dots) all events over the threshold of 35 knot along one line, an example of a data series that is strongly clustered in time. Clustering is due to (i) seasonal effects (cyclicity) and (ii) fewer Atlantic Basin events occurring (on average) during El Niño years (e.g. April 1986–March 1988) and more events occurring (on average) during La Niña years (e.g. April 1988–June 1989).



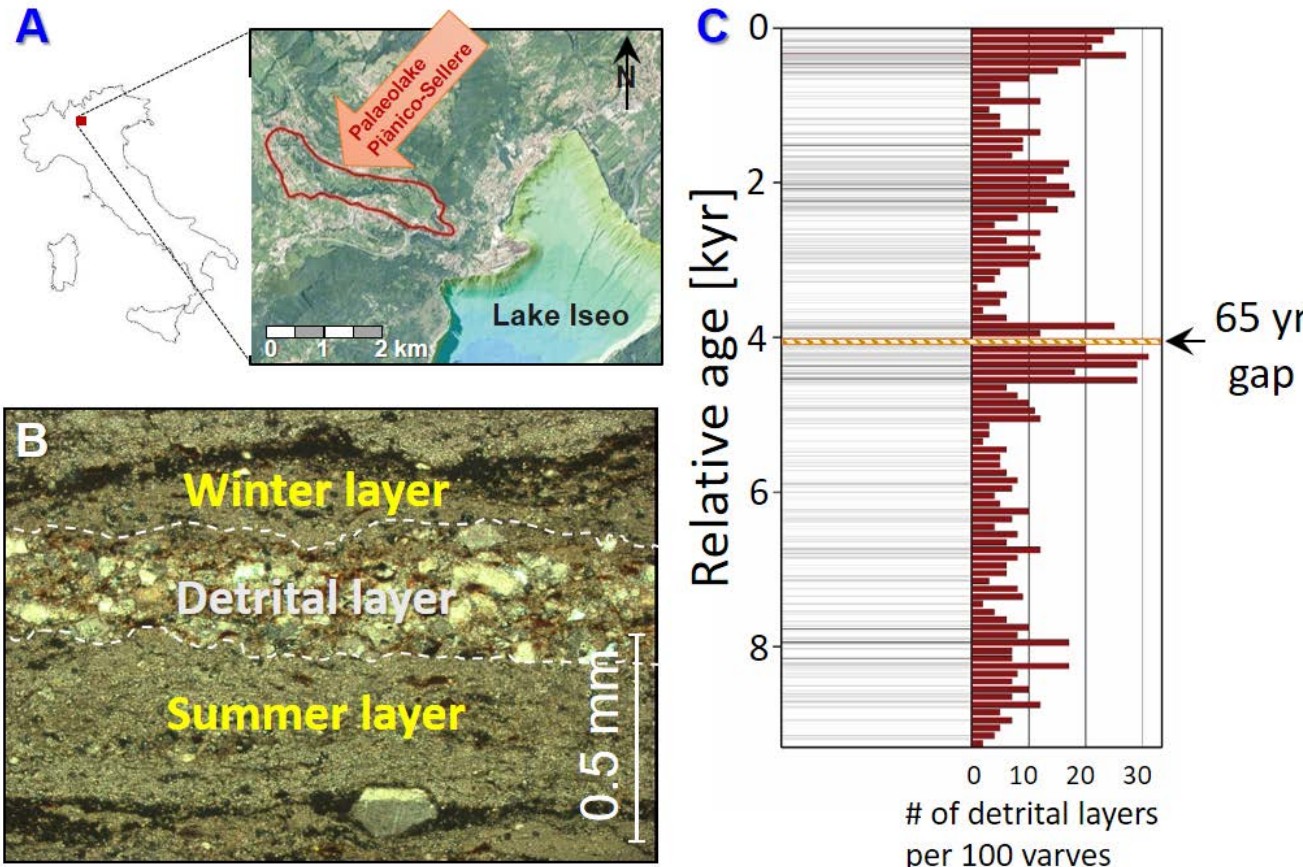

**Figure 2: Site, sediment outcrop and data.** (A) Location of the Palaeolake Piànico-Sèllere, Italy. (B) Detail of the 9.5 m sediment outcrop containing a winter and a summer layer with, shown here, a detrital layer in-between, that is interpreted to be a flood event. (C) Graphical representation of the sediment outcrop with the grey horizontal lines indicating detrital layers and the striped bar showing a 65 yr gap (see **Fig. 3**, for more detailed time series). The relative age represents years before the top of the stratigraphic section examined, with actual age estimated to be anywhere from 390 to 780 ka (see text).





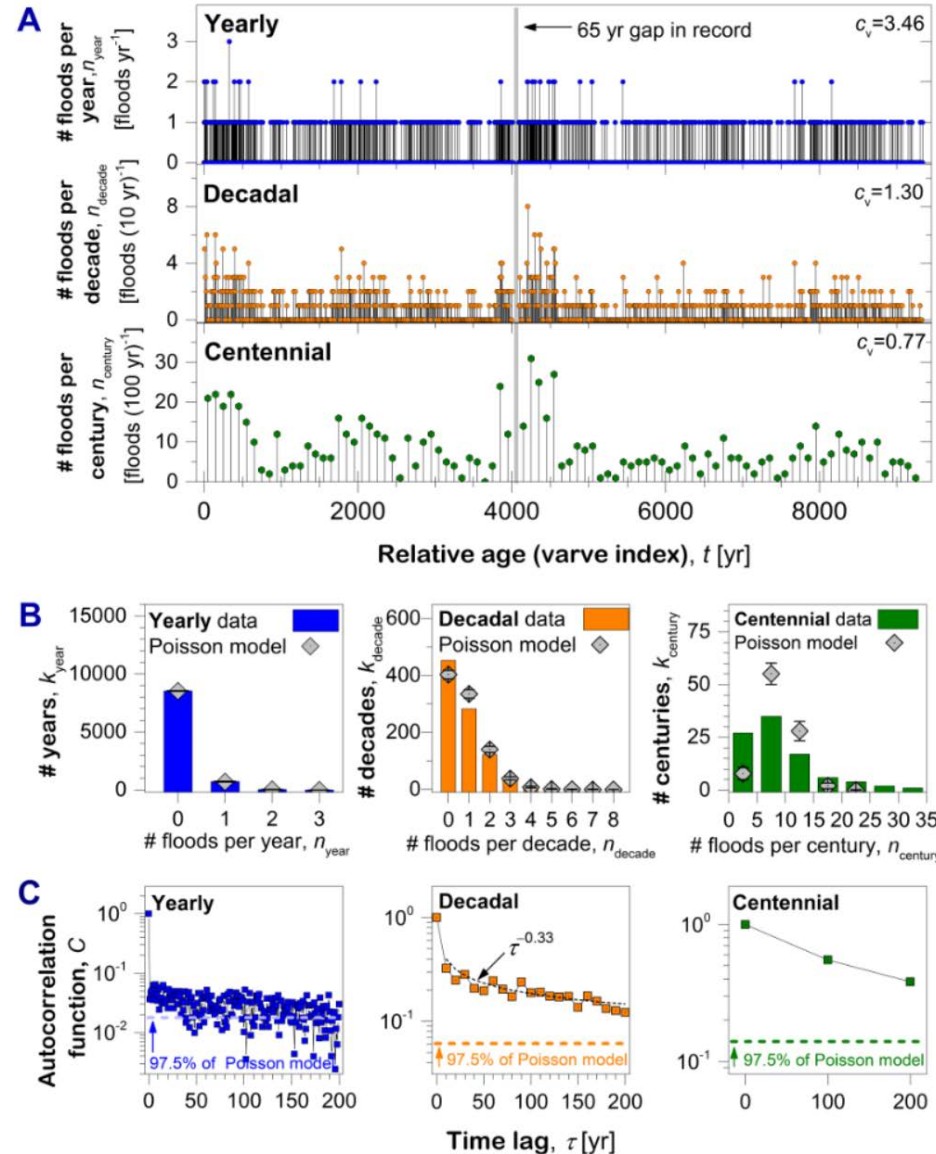

**Figure 3: Temporal succession, histograms and autocorrelation of the detrital layers of the 9336 yr Piànico-Sèllere, Italy palaeoflood record.** (**A**) The number of observed detrital (flood) layers per year (blue dots, $0 \leq n_{year} \leq 3$ floods $yr^{-1}$), decade (orange dots, $0 \leq n_{decade} \leq 8$ floods $decade^{-1}$) and century (green dots, $0 \leq n_{century} \leq 31$ floods $century^{-1}$) are presented over the 9336 yr of record examined. The x-axis represents relative age, with 0 yr the most recent varve and increasing values indicating further back in time. The grey bar represents a sediment gap of 65 yr (leaving 9271 yr of record examined over the 9336 yr). In addition, for each time resolution is given (upper right) the coefficient of variation $c_V = \sigma/\mu$, where $\sigma$ is the standard deviation and $\mu$ is the mean of the given time series. (**B**) Histogram of the number floods per year (blue bars), decade (orange bars) and century (green bars), with each compared to the Poisson model with respective rate parameters $\lambda_{year} = 0.083$ floods $yr^{-1}$, $\lambda_{decade} = 0.83$ floods $decade^{-1}$, $\lambda_{century} = 8.3$ floods $century^{-1}$. Diamonds represent the mean Poisson model value over 100 realisations, and the error bars the standard deviation of the realisations. (**C**) Autocorrelation function (ACF) of the number of floods per year (blue squares), decade (orange squares) and century (green squares). Also shown for the ACF of the number of floods per decade, is the best-fitting power-law model to the ACF (black dotted line). Also shown are the 97.5 percentile, i.e., the upper bound of the 95% confidence interval of the ACF (for lags $\tau > 0$) of an uncorrelated signal with the same one-point probability distribution.



**Figure 4: Data series of detrital layers (floods) and interevent occurrence times (IEOTs) from the Piànico-Sèllere, Italy palaeoflood sequence. (A)** In (A.1) are shown the number of detrital layers (floods) per varve, $n_{year}$, given as a function of varve number $t = 1$ to 9336 yr (blue dots) as shown in **Fig. 3**. In (A.2) a 100 yr portion of (A.1) (from $t = 8090$ to 8190 yr) is expanded with an illustration of four flood years that have detrital layers (floods) in them, and the interevent occurrence times $\Delta$ between them. These four flood years, each with $n_{year} \geq 1$ flood in them, occur as yellow symbols in all panels. **(B)** Interevent occurrence times, $\Delta$, temporally located (i.e., as function of the relative age) when they occurred. **(C)** Interevent occurrence times, $\Delta_j$, plotted as function of 'natural' time, where each $\Delta$ is no longer represented temporally when it occurs, but rather successively one after another, $j = 1, 2, 3,..., 741$; interevent occurrence times of $\Delta_j = 0$ are not shown.



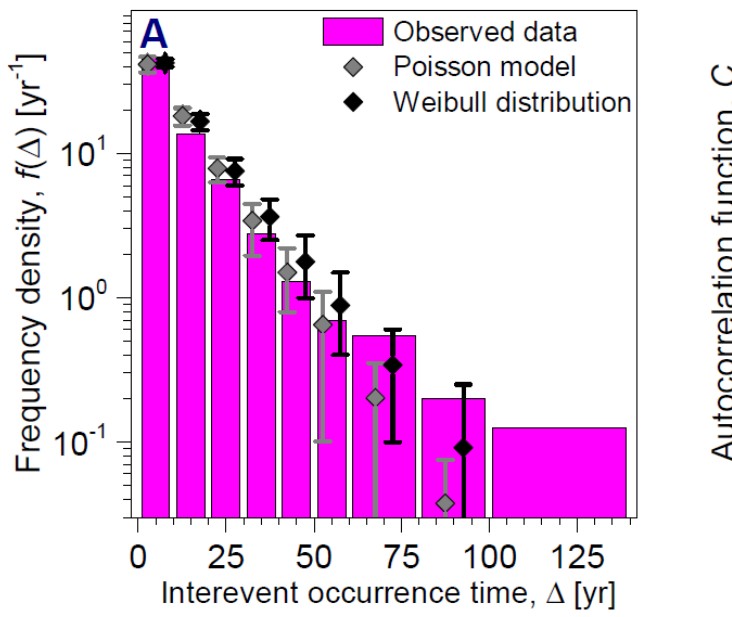

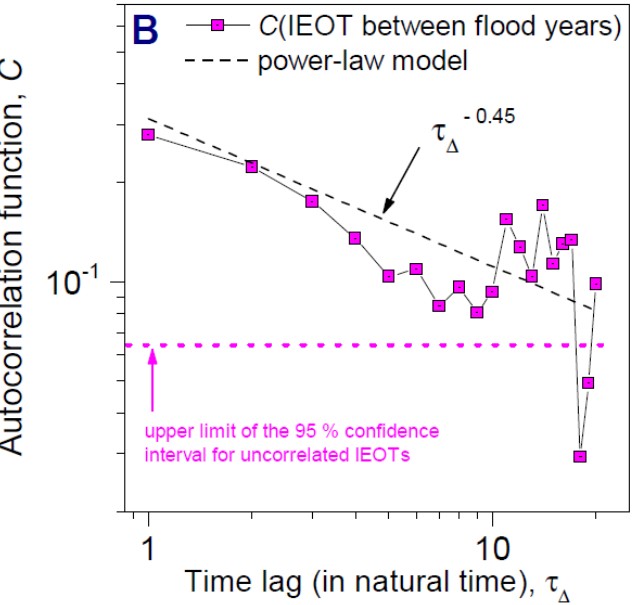

**Figure 5: Frequency-size and autocorrelation analysis of the interevent occurrence times (IEOTs) and comparison to the Poisson model.** (A) Frequency density of the IEOTs between the flood years given as a function of IEOTs in yr (pink vertical bars, semi-logarithmic scale), the corresponding best-fit Poisson model (grey diamonds) which was considered in **Fig. 3B** and the best-fit Weibull distribution (black diamonds), both with 95% confidence intervals. (B) Autocorrelation function of the IEOTs between flood years (those years with $n \geq 1$ flood). Also given is the best fitting power-law model (dashed black line) and the upper limit of the 95% significance for the autocorrelation function of a non-correlated time series with the same one-point probability distribution (horizontal dotted line, pink).





**Figure 6: Examples of synthetic fractional Gaussian noises with different modelled strengths of long-range persistence, $0.0 \leq \beta \leq 1.0$.** The presented data series, which have 512 elements each, are normalized to have a mean of zero and a standard deviation of one, and were created by Fourier filtering (see Appendices 1 and 2 in Witt and Malamud, 2013 for further details).





**Figure 7. Detrended Fluctuation Analysis (DFA) of the 9271 yr Piànico-Sèllere, Italy palaeoflood record**: The fluctuation function $F$ for the number of floods per decade shown as a function of the segment length $l$ and for different orders of the detrending (see legend) on a double logarithmic scale. Segments containing parts of or the entire gap were excluded. Also shown are best fitting power-law function for DFA3 and the corresponding power-law exponent $\alpha$ and its corresponding value of $\beta_{DFA} = 2\alpha - 1$.

.





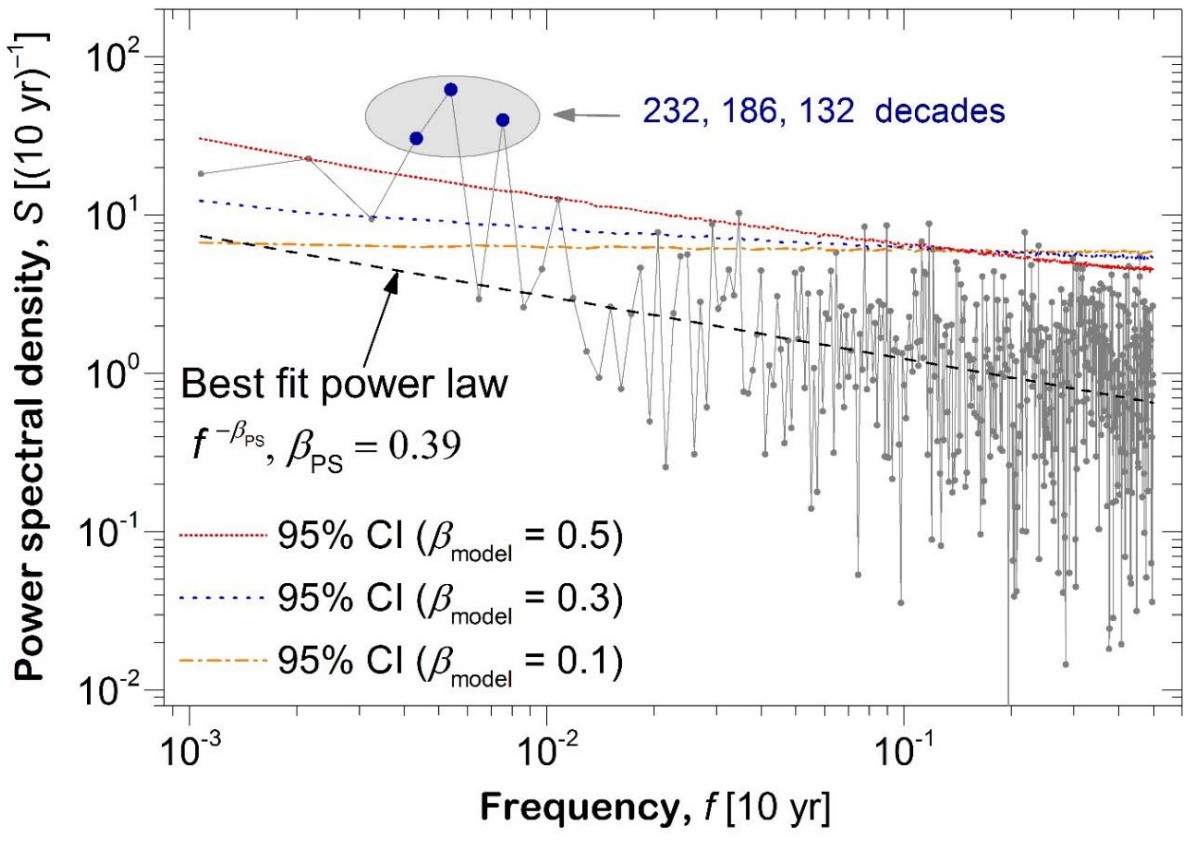

**Figure 8: Power spectral analysis:** The power spectral density (periodogram) is presented as a function of the frequency (measured per decade) for the number of floods per decade. Also given is the best fitting power-law function and the corresponding power-law exponent. Additionally, presented are the 95th percentiles of the power spectra created for synthetic data based on peaks over thresholds of fractional noises (see **Sect. 4.1**) for different strength of persistence (see legend, and description in text). Highlighted are the three values of the power spectral density that exceed all of these 95% percentiles, the corresponding cycle lengths are given.





**Figure 9: Analysis of flood rate fluctuations: (A)** Shown is the time dependent flood rate $\Lambda$ (black line) and its 95% confidence bands (green bands) as a function of the relative age $t = 1$ to 9336 yr for the palaeoflood data set shown in **Fig. 3A** which was computed by using a kernel density estimator. The grey colour indicates results that might be affected by boundary effects. The estimator is based on a Gaussian kernel with a width of $\pm 3\sigma$, where $\sigma = 250$ yr (**Appendix B**), shown in the inset figure. Also shown is the mean flood rate (dashed blue line) of $\lambda_{year} = 0.83$ floods yr$^{-1}$. **(B)** For a comparison the number of floods per century $n_{century}$ (green bars) is presented. The grey vertical bar indicates the gap in the data set.



**Figure 10: Schematic of the POT$_{FGN+Period}$ model: (A)** Shown is a fractional Gaussian noise with a persistence strength of $\beta_{model} = 0.5$. The time resolution is $\Delta t = 1$ y. **(B)** Three thresholds are set (dashed horizontal lines). All noise values that are below the threshold $\theta_{01}$ are considered as years with zero floods, all noise values between the thresholds $\theta_{01}$ and $\theta_{12}$ [$\theta_{12}$ and $\theta_{23}$] are considered as years with one [two] floods, and all noise values that exceed the threshold $\theta_{23}$ are considered years with 3 floods. The thresholds $\theta_{01}$, $\theta_{12}$ and $\theta_{23}$ are chosen such that the resultant one-point probability distribution is the same as our flood series **(C)** Time series of the modelled number of floods per year (blue circles) for 1000 years. For parts (A), (B) and (C) are also shown the histogram of the number of values at a given size.




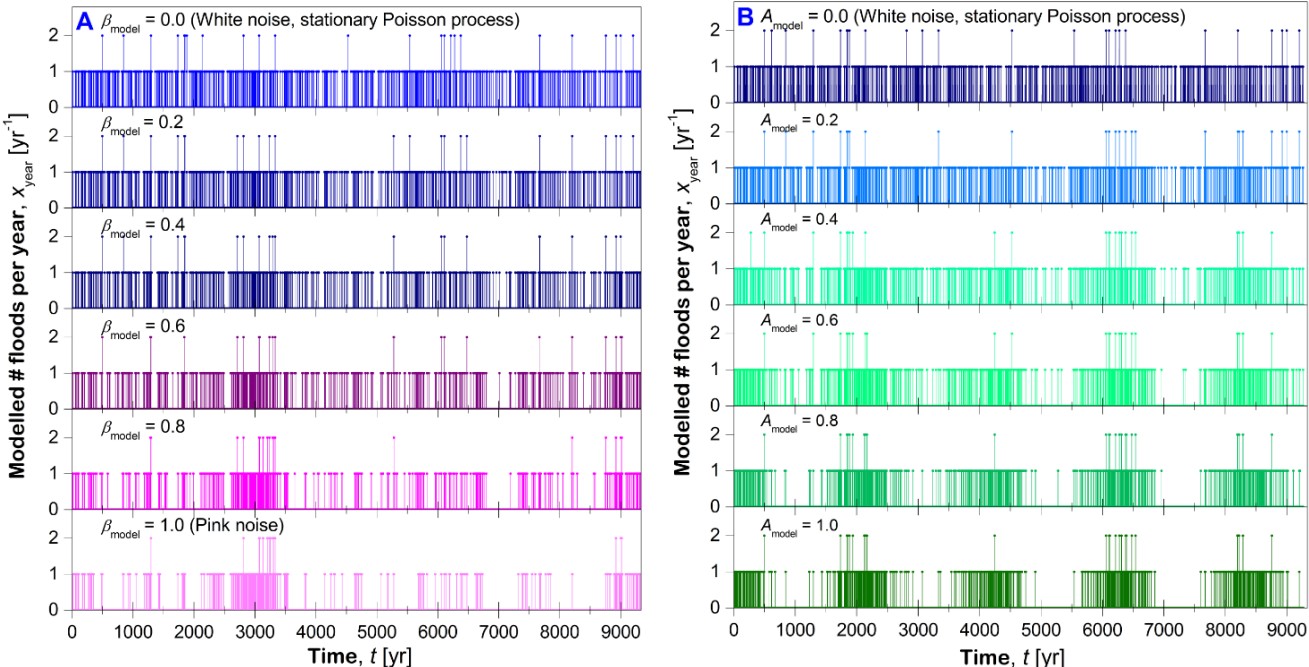

**Figure 11: POT$_{\text{FGN+Period}}$ model realisations:** Shown are **(A)** long-range persistent flood time series that are modelled with the proposed POT$_{\text{FGN+Period}}$ model (see **Fig. 10**) as peaks over thresholds (POT) of a fractional Gaussian noise (illustrated in **Fig. 6**) with different persistence strengths $\beta_{\text{model}} = 0.0, 0.2, 0.4, 0.6, 1.0$, and **(B)** flood time series $x_{\text{year}}$ with a long-term periodic cyclicity which are modelled as peaks over thresholds (POT) of a Gaussian white noise ($\beta_{\text{model}} = 0.0$) superimposed with a sine wave ($T = 2000$ yr) with different sine wave amplitudes $A_{\text{model}} = 0.0, 0.2, 0.4, 0.6, 1.0$.





**Figure 12: Analysis of model realisations of the POT$_{FGN+Period}$ model with different pairs of parameters for persistence strength and amplitude of the periodic component**. Given for all six panels are the two model parameters of the POT$_{FGN+Period}$: (i) *strength of long-range persistence* of the underlining fractional Gaussian noise $\beta_{model} = 0.00$ to $1.00$ (step size 0.02) on the *x*-axis and (ii) the *amplitude of the superimposed periodic component* (period $T = 2030$ yr), $A_{model} = 0.00$ to $1.00$ (step size 0.02) on the *y*-axis. For each of the $51 \times 51 = 2601$ pairs of model parameters ($\beta_{model}$, $A_{model}$) 1000 realisations were produced. Each realisation is a synthetic floods per year series which is constructed as described in **Sect. 4.1** and illustrated in **Fig. 10**. In **A** to **D** are given the mean of the measured $\beta_{PS}$, $\beta_{DFA}$, $k_W$ and $A_{rate}$ (see



legend for colours and contours) for the 1000 realisations in each of the $51 \times 51$ cell parameter pairs ($\beta_{\text{model}}$, $A_{\text{model}}$). In panel **E**, we then take *measured* values from each of the panels **A** to **D** and show in Panel **E** their corresponding ($\beta_{\text{model}}$, $A_{\text{model}}$) values. The measured values are chosen from each panel to be the same range (with error bars) as our Piànico palaeoflood data as follows: (i) strength of persistence, $0.370 \leq \beta_{\text{PS}} \leq 0.410$, 5000 values from Panel **A**, magenta dots; (ii) strength of persistence, $0.23 \leq \beta_{\text{DFA}} \leq 0.27$, 5000 values from Panel **B**, brown dots; (iii) shape parameter of the Weibull distribution, $0.77 \leq k_{\text{W}} \leq 0.81$, 5000 values from Panel **C**, blue dots; and (iv) sinusoidal flood rate, $0.0450 \leq A_{\text{rate}} \leq 0.0530$, 5000 values from Panel **D**, green dots. When a given realisation has $\beta_{\text{PS}}$, $\beta_{\text{DFA}}$, $k_{\text{W}}$ and $A_{\text{rate}}$ all that satisfy the preceding conditions (i.e., all four colours for a given realisation appear in Panel **E**), then we give the colour a black dot. In panel **F**, the 2D probability density of model parameters ($\beta_{\text{model}}$, $A_{\text{model}}$) are shown whose persistence strength ($\beta_{\text{DFA}}$), shape parameter of the Weibull distribution ($k_{\text{W}}$) and best fitting sinusoidal flood rate have similar values to that of the Piànico palaeoflood record (see legend for the colour code). Marked is the mode (red bullet) and the area that contains approximately 50% of the points (blue dashed lines).





**Figure 13: POT$_{FGN+Period}$ model with optimized parameters: (A)** Shown (black time series) is a realisation of a fractional Gaussian noise (FGN) with a long-range persistence strength of $\beta_{model} = 0.24$ (normalized to a mean $= 0.0$ and variance $= 1.0$) that is superimposed with a periodic component (period: $T = 2030$ yr, amplitude $A = 0.30$). This synthetic time series is given a time resolution of $\Delta t = 1$ yr and the length $N_{year} = 9270$ yr. Peaks over thresholds (POT) are considered (as in **Fig. 10**) for five thresholds (three shown here) $\theta_{01}$, $\theta_{12}$, etc. where input (FGN+period) values $< \theta_{01}$ translate to $x_{year} = 0$ flood yr$^{-1}$, input values between $\theta_{01}$ and $\theta_{12}$ result in $x_{year} = 1$ flood yr$^{-1}$, etc. The resultant model series of the number of floods per year (grey and blue circles) for 9270 years is constructed to have a Poissonian one-point distribution and is shown at the bottom of the panel **A**. **(B)** Shown are 50 more model realisations of the POT$_{FGN+Period}$ model as number of floods per year (see legend for the colour code), with maximum $x_{year} = 4$ floods yr$^{-1}$. **(C)** Time dependent flood rate $\Lambda(t)$ given as a function of time $t$



(see **Fig. 9A**, text and **Appendix B**) using 1000 model realisations similar to Panel **B**. We first compute a kernel density estimator for each of the 1000 model realisations. Then we take the mean of these at each time step, giving us a mean $\Lambda(t)$ (dashed black line). We then compute the 95% confidence band limits (dark grey bands), by ordering the 1000 values of $\Lambda(t)$ for each time step $t$, and choosing the $25^{th}$ and $975^{th}$ values. As the flood rate of the first and last 500 years might be affected by edge effects, the corresponding dark grey colour banding is changed to light grey. For a comparison the corresponding time dependent flood rate for the palaeoflood data (solid black line), as shown in **Fig. 9A**, is given. **(D)** Black diamonds represent the mean number of floods per year, decade and century based on 1000 realisations of our POT$_{FGN+Period}$ model as given in the previous panels, with error bars 95% confidence intervals. For comparison are shown: (i) histogram (also shown in **Fig. 3**) of the original palaeoflood time series number of floods per year (blue bars), decade (orange bars) and century (green bars); (ii) the corresponding Poisson model (grey diamonds, also shown in **Fig. 3**) with respective rate parameters $\lambda_{year} = 0.083$ floods yr$^{-1}$, $\lambda_{decade}$ = 0.83 floods decade$^{-1}$, $\lambda_{century} = 8.3$ floods century$^{-1}$.





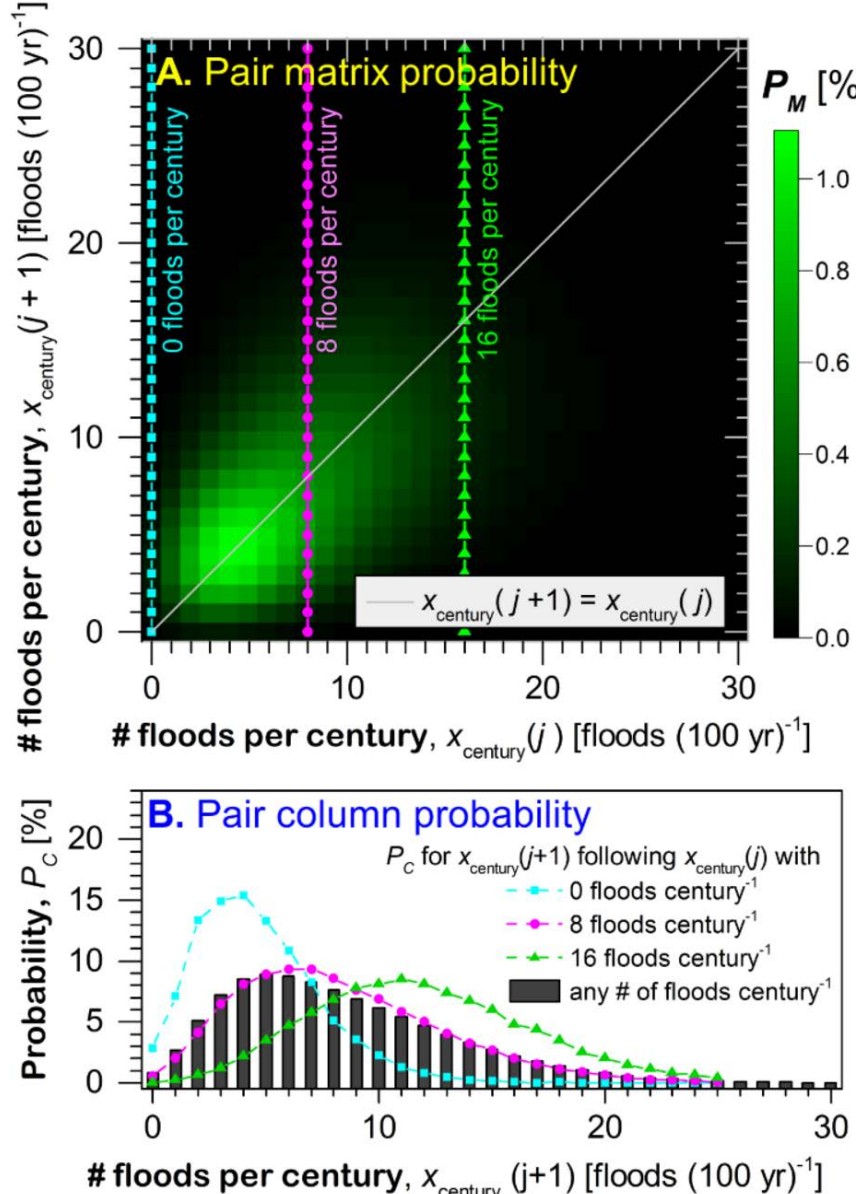

**Figure 14: Using the optimized POT$_{FGN+Period}$ model for forecasting the number of floods in one century based on the number of floods in the previous century:** Based on the palaeoflood model realizations for one billion ($10^9$) years are shown **(A)** The 2D probability of the number of floods per century ($x_{century}(j)$) and the number of floods in the succeeding century ($x_{century}(j+1)$) (background grid colours, see legend for scale). The solid grey diagonal line represents equality of the number of floods in both centuries ($x_{century}(j+1) = x_{century}(j)$). **(B)** The two-dimensional probability distribution shown in (A) is cut vertically at $x_{century} = 0$, 8 and 16 floods century$^{-1}$ and the corresponding probability densities of floods per centuries after a century with $x_{century} = 0$, 8 and 16 floods century$^{-1}$ are presented as coloured lines (see legend). Also given is the probability density of floods per century of the palaeoflood model.