# Peer review of "Analysis and Modelling of a 9.3 kyr Palaeoflood Record: Correlations, Clustering and Cycles"

_Hydrology and Earth System Sciences, 2016_

## Referee Comment (RC1) · Anonymous Referee #1 · 22 Dec 2016

This was a thorough and interesting paper to read.

It is regrading relatively novel, but important topics in palaeoscience. The degree to which we can make informed interpretations of proxy data is constrained by our understanding of the memory/persistence of a given proxy in a given archive. Finding statistical approaches to help with this understanding is therefore important (e.g. Fischer, 2016).

This not only helps with interpretations of time-series of past change but also in understanding how best to statistically model future scenarios, including the frequency of hazards such as flooding.

It draws heavily on Witt and Malamud (2013) and applies their approaches to a varved lake record. Links to this prior paper, and the useful R code it contains, are clear in this

paper and with the flood record to also be made available, allow readers to undertake the analyses and check output before applying to their own data. This is likely to be a very useful product for the community, particularly for working with annual time series such as those available from varved lake sequences.

Please note, I haven't had time to work through this for the review.

There are a couple of small typos in the text, but this is a paper that is publishable as it is.

Fischer, M.J., 2016. Predictable components in global speleothem $\delta$18O. Quat. Sci. Rev. 131, Part B, 380-392. Witt, A., Malamud, B.D., 2013. Quantification of Long-Range Persistence in Geophysical Time Series: Conventional and Benchmark-Based Improvement Techniques. Surveys in Geophysics 34, 541-651.

---

## Author Comment (AC1) · 16 Feb 2017

We thank the reviewer for their positive comments and will examine the manuscript for typographical errors in the next version.

---

## Referee Comment (RC2) · E. Zehe (Referee) · 31 Mar 2017

Summary: The authors analyze an impressive record of paleo floods, for clustering and long term autocorrelation of events at different temporal aggregation levels. To this end they use the presence of gravel enriched strata (detrital layers) as indicator for channelized inflow due to a flood into the lake. This seems plausible as those events might be associated with bedload transport and sedimentation into a reconstructed paleo lake in the Italian Alps. Data were collected at on sediment outcrop yielding a time series of nearly 10 000 y including proxies of 771 flood events.

The authors compare the histogram and the autocorrelation function of the data at different temporal aggregation levels of years, decades and centuries using a Poisson process as a Null model and found clear autocorrelations and thus memory at among

flood intensities within decades and centuries. They relate furthermore the average inter flood periods in the data set again to the Poission process as Null model. Last not least they propose and successfully test a stochastical model to simulate paleo flood time series based on the superposition of fractional Gaussian noise with a long term memory component.

Evaluation: The study deals with a highly interesting topic and is based on a sound data set. The presented results are very interesting and partly reveal long term memory at the decadal and centennial resolution, and suggest that floods cluster in time and following a Weibull distribution. The proposed study is hence without doubt very much worth to be published in HESS. Yet I think that the particularly the manuscript needs to be restructured and streamlined and parts of the presented analysis net a more critical reflection. I thus recommend the author should revise their study addressing the following points.

Main Points: - The presentation of the manuscript is in many parts a little too wordy. E.g. there is no need to explain the concept of correlation in such a depth in the intro, as the auto correlation within a time series is well known. Unfortunately the authors mix the presentation of the statistical methods and results, this makes the manuscript difficult to follow and to evaluate the partly very interesting findings. I suggest to rearrange the presentation in a more old school manner and to present methods, results and discussion of those in separate sections.

- I wonder why the authors do not employ variograms/semivariances to estimate the spatial correlation time. This will better separate the un-correlated from the correlated variability in their records. In this context I wonder, how much of the increase in correlation times at the hifher aggregation levels may be attributed to well known scaling effects arising form to the aggregation process itself. The range of variogram estimated from data points increased when these are aggregated to blocks (or time intervals here) while nugget increases and sill decreases – this is well known from variogram regularization. This effect could be easily determined using data from your Poisson null model.

[Figure]

- I appreciate that one prefers the use of power law functions with infinite correlation length – but what is the physical meaning of this?

Minor points: - The authors use the presence of gravel enriched strata as indicator for channelized inflow due to a flood into the lake. This seems plausible as those events might be associated with bedload transport and sedimentation into the lake. However, there is no linear relation between flow velocity and bed load transport capacity. The literature is full of empirical formulas for this. This implies that the thickness of these strata is not linearly related to pealk discharge (flow velocities). The authors might consider this as an additional argument against the use of layer thickness as proxy for flood peak discharge.

- Why not showing a long time series of annual flood maxima, they also exhibit cluster-ing of extremes and long time periods – this example is maybe closer to the object of desire?

- Could you also infer on clustering of intensities above a threshold using indicator auto correlations?

- I appreciate the authors effort to provide a reproducible paper, but the explanation of the Poisson process is too detailed, this is text book knowledge. The parameter lambda is the mean value of floods / time interval, which depends on the aggregation level. As far as I know the Poisson process produces cluster data, if lambda is larger than 1?

- Page 9 line 5: In case the flood do always occur in the first three years of the decade the average inter event period is not 1 year but, 2/3 *1 + 1/3*7= 3 years?

Best regards,

Erwin Zehe

---

## Author Comment (AC2) · 8 May 2017

**Reply to Erwin Zehe on "Analysis and Modelling of a 9.3 kyr Palaeoflood Record: Correlations, Clustering and Cycles" by Witt *et al.* (2017)**

We thank Erwin Zehe (EZ) for his thoughtful comments on our manuscript hess-2016-470 (Witt *et al.*, 2017). In this revision of our manuscript, we address the reviewer comments as given below, and believe that the result will be an improved manuscript.

**EZ MAJOR COMMENT 1**. "The presentation of the manuscript is in many parts a little too wordy. E.g. there is no need to explain the concept of correlation in such a depth in the intro, as the auto correlation within a time series is well known. Unfortunately the authors mix the presentation of the statistical methods and results, this makes the manuscript difficult to follow and to evaluate the partly very interesting findings. I suggest to rearrange the presentation in a more old school manner and to present methods, results and discussion of those in separate sections."

**Author reply**
(i) **Wordiness and detail of explanation.** We agree that this is a long paper (but we hope well organized, so people can dip in and out of certain sections), and that in places the manuscript tends towards facts/information that might be well known by certain parts of the community. We will relook at the entire manuscript, editing words out where possible, but also maintaining a level of readability and accessibility by those who do not have a background in statistical methods related to correlations, clustering, and cyclicity. We will also examine whether parts of the manuscript should be moved towards an appendix.

(ii) **Mixture of statistical methods and results. Rearrange manuscript to be methods, results, discussion**. We propose to take all statistics 'methodology' sections in Section 3, pull these out into a new section. We would then have as outline the following:

1.0  Introduction
2.0  Data
3.0  Definitions and methods for the analysis of event series
      3.1 Poisson process
      3.2 Autocorrelation
      3.3 Interevent occurrence time probability distributions
      3.4 Long range correlations using detrended fluctuation analysis and power-spectral analysis
4.0  Results of analysis of the palaeoflood time series
      4.1 Number of palaeofloods per year, decade and century, compared to a Poisson process
      4.2 Temporal correlations using autocorrelation
      4.3 Clustering using interevent occurrence times
      4.4 Quantifying long-range correlations using DFA and power-spectral analysis
      4.5 Cyclicity and fluctuations in the 9.3 kyr palaeoflood record
      4.6 Summary of clustering, correlation and cyclicity of the 9.3 kyr Piànico-Sèllere palaeoflood record
5.0  Creating a peaks over threshold model to capture correlations and clustering of the observational data
      *[Sections 5.1, 5.2, … the same as 4.1, 4.2, … previously]*
6.0  Summary

**EZ MAJOR COMMENT 2** "I wonder why the authors do not employ variograms/semivariances to estimate the spatial correlation time. This will better separate the un-correlated from the correlated variability in their records. In this context I wonder, how much of the increase in correlation times at the hifher aggregation levels may be attributed to well-known scaling effects arising form to the aggregation process itself. The range of variogram estimated from data points increased when these are aggregated to blocks (or time intervals here) while nugget increases and sill decreases – this is well

known from variogram regularization. This effect could be easily determined using data from your Poisson null model."

**Author reply:**

The reviewer has suggested that we use semivariograms as a model for finite range correlations or short-range persistence (i.e., a temporal correlation time), where at a given lag or less in our time series, correlations are observed, but that at that lag or greater, correlations are not observed, or the correlations fall below a statistical threshold. The reviewer further then suggests that any increase in the correlation time (based on short-range persistence), might be due to the scaling effects from the aggregation process itself.

In our manuscript, we have used autocorrelation as a way of first exploring temporal correlations, and we found up to 200 years lags there were correlations (see also reply to Comment #3). In other words, we did not identify (up to 200 years) a temporal correlation time. We agree that we should go farther in our temporal lags, and will do so in the next version of the manuscript.

Autocorrelation is linked to the semivariogram, but includes the mean of the time series. We will also do a semivariogram analysis to see if a short-range correlation model might also be appropriate within statistical significance.

We also note that in our manuscript after our initial explorations of the time series using autocorrelation, we decided to apply a model of long-range correlations (i.e., one that does not have a temporal correlation time). Certainly, semivariograms can be used a long-range persistence model. However, instead of semivariogram analysis we applied Detrended Fluctuation Analysis (DFA) and Power-Spectral analysis (PS) for quantifying the long-range persistence strength of our palaeoflood and synthetic time series. Both semi-variogram analysis and DFA operate in the time domain, and power-spectral analysis in the frequency domain. In research we did four years ago and published as a review paper (Witt & Malamud, 2013) we examined systematically, using 17,000 synthetic benchmark series, the types of errors when evaluating long-range persistence using these three techniques. All three are appropriate measures for long-range persistence; however, the systematic and random errors (biases and standard errors) of DFA and PS analysis are significantly smaller (particularly for time series with non-Gaussian one-point probability distribution, and with very few values in the time series) compared to semi-variogram analysis, and both DFA and PS analysis are appropriate for a much broader range of long-range persistence strengths compared to semi-variogram analysis. We put our research into the context of the wider literature which has found similar results. We therefore in this paper use DFA and power-spectral analysis, due to it being much more robust than semivariograms as a long-range model. We will add to our manuscript a short explanation why we have chosen DFA and power-spectral analysis (compared to semivariograms) for examining long-range persistence.

**EZ MAJOR COMMENT 3** "I appreciate that one prefers the use of power law functions with infinite correlation length – but what is the physical meaning of this?"

**Author reply:** We believe (as this refers to correlation length) this comment refers to our presentation of the auto-correlation function applied to our palaeoflood time series (per year, per decade, per century), and the auto-correlation decaying as an inverse power-law with respect to the temporal lag. We will take "correlation length" here as meaning the lag at which values are separated and still statistically correlated. One could also consider power-spectral analysis and DFA, in terms of an equivalent correlation length for which values are still, on average, statistically correlated. Self-affine long-range memory is defined to be a time series where 'all' values in the time series are on average statistically correlated to one another. In the case of long-range memory the correlation length is defined to be infinite, due to the power-law decay of the autocorrelation function as a function of the temporal lag. The first physical significance of this 'infinite' correlation length is that 'all' values are

statistical correlated with one another in the time series. A second physical significance, is that autocorrelation functions with a power-law decay indicate a self-affine geometry in the original time series examined, i.e., the considered object can be explained as a stochastic fractal. In the case of our palaeoflood series this means that the floods are organised in time as the points of a Cantor dust organised in space. We will add some sentences about this aspect to our manuscript.

**EZ MINOR COMMENT 1**: "The authors use the presence of gravel enriched strata as indicator for channelized inflow due to a flood into the lake. This seems plausible as those events might be associated with bedload transport and sedimentation into the lake. However, there is no linear relation between flow velocity and bed load transport capacity. The literature is full of empirical formulas for this. This implies that the thickness of these strata is not linearly related to peak discharge (flow velocities). The authors might consider this as an additional argument against the use of layer thickness as proxy for flood peak discharge."
**Author reply**: We agree that strata (layer) thickness is not an indicator for flood magnitude. Note that in our detrital layers, which represent the flood events, the detrital layers themselves are mainly constituted of dolomite from the catchment, i.e. fine grained sediment with no gravel. We will add (in yellow below) a short statement to our original manuscript to clarify.

"Although detrital layers, ==which are mainly constituted of dolomite==, indicate the occurrence (year) of an extreme precipitation event (flood), the thickness is not taken as representative of the magnitude of the event, as the thickness of a layer may depend on the distance between the source of the detritus and the studied section, the amount of detritus available at the time of the extreme precipitation and weaker vs. stronger rainfall during the extreme precipitation events. In addition, over the 9336 yr period we investigate, the source of given storms might result in different thickness signatures. We also note that although we can identify when a flood is thought to have occurred, this does not preclude the possibility that other floods occurred but were not identified as a detrital layer. …"

**EZ MINOR COMMENT 2**: "Why not showing a long time series of annual flood maxima, they also exhibit clustering of extremes and long time periods – this example is maybe closer to the object of desire?"
**Author reply:** If we understand what is being suggested, is that we replace parts C and D of Figure 1 in our manuscript (Atlantic Storm Maximum Wind speed) with a similar figure related to floods. We probably would not use an annual flood maxima time series, as one year might have a flood that occurs that is less than three 'independent' floods in the next year. Rather, we would use previous techniques that we have worked with to do partial duration flood series, and then choose the largest 100 for the period considered. We agree this could make the Figure 1 more relevant to our paper.

**EZ MINOR COMMENT 3**: "Could you also infer on clustering of intensities above a threshold using indicator auto correlations?"
**Author reply:** We understand this question to ask what would happen if we applied the autocorrelation function to our palaeoflood time series (per year, per decade, per century) to all values above a given threshold, in terms of learning more about the clustering (vs. correlation) behaviour of the time series. The values above the threshold could either (time series 1) retain their value, and all values below the threshold set to '0', or (time series 2) all values above the threshold set to a given value (e.g., '1') (and all values below the threshold '0').

Currently we apply autocorrelation analysis to the number of floods per year, per decade, and per century, but without a threshold. The problem we have with our palaeoflood time series, is that rather a time series that is 'just' stochastic, there is also the presence of a long-term cyclicity. In case of weakly-stationary data such as a fractional noise (no underlining long term cycle), we could use the

decay rate of the autocorrelation function for quantifying correlations (and also examine values above the threshold to infer clustering behaviour).

However, because of the presence of a long-term cyclicity, the autocorrelation function will be strongly impacted, and we therefore use Detrended Fluctuation Analysis (DFA) and power-spectral analysis, which are more appropriate for fractional noises with cyclicity. For example, DFA is detrending segments of the aggregated time series.

In our manuscript, to examine clustering, we use the one point probability distribution (and in particular the shape parameter of the Weibull distribution) of the time intervals between floods as an indicator for clustering. For our specific model, small shape parameters are equivalent to a large strength of long-range correlations and vice versa (see Figure 12 a,b and c) and thus strong clusters appear in case of strong long-range correlations.

We will add in a sentence to our manuscript to refer to the fact why we do not examine autocorrelation as an indicator of clustering.

**EZ MINOR COMMENT 4**: "I appreciate the authors effort to provide a reproducible paper, but the explanation of the Poisson process is too detailed, this is text book knowledge. The parameter lambda is the mean value of floods / time interval, which depends on the aggregation level. As far as I know the Poisson process produces cluster data, if lambda is larger than 1?"
**Author reply:**
(a) We will shorten the introduction of Poisson processes in our introduction, but would still like enough detail that a non-expert will have enough to follow the flow of our manuscript.
(b) The Poisson process as introduced in our original manuscript page 8, does not lead to temporal clustering of the realisation (in time) of a Poisson process, for any lambda > 0 (Cox and Lewis, 1978). We also note that for lambda > 1 the one-point probability distribution of events per time unit is one humped and has a defined mode.

**EZ MINOR COMMENT 5**: "Page 9 line 5: In case the flood do always occur in the first three years of the decade the average inter event period is not 1 year but, 2/3 *1 + 1/3*7= 3 years?"
**Author reply**: We thank the reviewer for noting this error. We will change
"For example, if in a given decade, we have three varves each with one flood, Eq. (2) will be different if the three floods are the first three years of the decade (i.e., mean $\Delta$ = 1 yr) vs. a flood in years 1, 5, and 9 of the decade (i.e., mean $\Delta$ = 4 yr)."
to read
"For example, if in a given decade, we have three varves each with one flood, Eq. (2) will be different if the three years with floods are in years 1, 2, 3 ($\Delta_1 = \Delta_2$ = 1 yr, $\Delta_3 \geq 8$ yr because the 4$^{th}$ flood will be in a subsequent decade) vs. one flood each in years 1, 5, and 9 of the decade (i.e., $\Delta_1 = \Delta_2$ = 4 yr, $\Delta_3 \geq 2$ yr)."

References Cited
Cox, D.R., and Lewis, P.A.W.: *The Statistical Analysis of Series of Events*, Chapman and Hall, London, 1978.
Witt, A., and Malamud, B.D.: Quantification of long-range persistence in geophysical time series: Conventional and benchmark-based improvement techniques. *Surveys in Geophysics* **34** 541–651, 2013.

---

## Author Response (AR1)

**Reply to Reviewers on "Analysis and Modelling of a 9.3 kyr Palaeoflood Record: Correlations, Clustering and Cycles" by Witt *et al.* (2017)**

We thank the editor, Anonymous Reviewer 1, and in particular Erwin Zehe (EZ) for their thoughtful comments on our manuscript hess-2016-470 (Witt *et al.*, 2017). We have extensively revised our manuscript mainly due to the many very constructive comments of Erwin Zehe.

As suggested by Erwin Zehe, we have reduced some of the text, reordered the manuscript (and figure numbering) as suggested so that the methods are now separate from the results—this is much more readable now, added two new sets of analyses and figures (extending autocorrelation function to greater lags, and doing a semivariogram of our data), and responded to his other thoughtful comments below and in the text. We believe that the result is much more readable with confusions over certain parts of this long paper now better clarified.

Below we reply to each of the reviewers. Attached we also include a track change document of what has been changed from the original manuscript.

**Reply to Anonymous Reviewer 1 on "Analysis and Modelling of a 9.3 kyr Palaeoflood Record: Correlations, Clustering and Cycles" by Witt *et al.* (2017)**

**Anonymous Reviewer 1 comments**. This was a thorough and interesting paper to read. It is regrading relatively novel, but important topics in palaeoscience.  The degree to which we can make informed interpretations of proxy data is constrained by our understanding of the memory/persistence of a given proxy in a given archive. Finding statistical approaches to help with this understanding is therefore important (e.g. Fischer, 2016). This not only helps with interpretations of time-series of past change but also in understanding how best to statistically model future scenarios, including the frequency of hazards such as flooding. It draws heavily on Witt and Malamud (2013) and applies their approaches to a varved lake record. Links to this prior paper, and the useful R code it contains, are clear in this paper and with the flood record to also be made available, allow readers to undertake the analyses and check output before applying to their own data. This is likely to be a very useful product for the community, particularly for working with annual time series such as those available from varved lake sequences. Please note, I haven't had time to work through this for the review. There are a couple of small typos in the text, but this is a paper that is publishable as it is.

**Author reply**
We thank the reviewer for their positive comments and all four of the authors have examined the manuscript for typographical errors in this previous draft, catching a number. We will take extra care at the Galley Proof stage also with the typos.

**Reply to Erwin Zehe on "Analysis and Modelling of a 9.3 kyr Palaeoflood Record: Correlations, Clustering and Cycles" by Witt et al. (2017)**

**EZ MAJOR COMMENT 1**. "The presentation of the manuscript is in many parts a little too wordy. E.g. there is no need to explain the concept of correlation in such a depth in the intro, as the auto correlation within a time series is well known. Unfortunately the authors mix the presentation of the statistical methods and results, this makes the manuscript difficult to follow and to evaluate the partly very interesting findings. I suggest to rearrange the presentation in a more old school manner and to present methods, results and discussion of those in separate sections."

**Author reply**

(i) **Wordiness and detail of explanation.** We agree that this is a long paper (but we hope well organized, so people can dip in and out of certain sections), and that in places the manuscript tends towards facts/information that might be well known by certain parts of the community. We have relooked at the entire manuscript, editing words out where possible, but also maintaining a level of readability and accessibility by those who do not have a background in statistical methods related to correlations, clustering, and cyclicity.

(ii) **Mixture of statistical methods and results. Rearrange manuscript to be methods, results, discussion**. We have now taken all statistics 'methodology' sections in the original manuscript Section 3, left them in Section 3, and then moved results to both Section 2 (data, where it was just describing the data) and Section 4 (analyses on clustering, correlations, and cyclicity). This was a tad time consuming to break up the methods and results, as they were heavily integrated before. We believe that the result is now much more readable, and thank the reviewer for their suggestion of this major change. Our outline is now as follows:

1.0   Introduction
2.0   Data
3.0   Definitions and methods for the analysis of event series
      3.1   Interevent Occurrence Times (IEOTs)
      3.2   Poisson processes as a model for an uncorrelated non-clustered time series
      3.3   Weibull distribution of IEOTs as an indicator for clustering
      3.4   Autocorrelation as a method for quantifying temporal (short-range and long-range) correlations
      3.5   Power-spectral analysis and detrended fluctuation analysis (DFA) for quantifying long-range correlations
      3.6   Fractional noises as examples of long-range correlated time series
4.0   Results of statistical metrics, clustering, correlations and cyclicity of the palaeoflood sequence
      4.1   Number of palaeofloods per year, decade and century, compared to a Poisson process
      4.2   Probability of interevent occurrence times (IEOTs) compared to a Weibull distribution as a potential indicator of clustering
      4.3   Autocorrelation of palaeoflood sequences to quantify correlations
      4.4   Autocorrelation of interevent occurrence times (IEOTs) as a potential indicator of clustering s
      4.5   Detrended fluctuation analysis (DFA) and power-spectral analysis for quantifying long-range correlations and as a potential indicator of cyclicity
      4.6   Cyclicity and fluctuations in the 9.3 kyr palaeoflood record
      4.7   Summary of clustering, correlation and cyclicity of the 9.3 kyr Piànico-Sèllere palaeoflood record
5.0   Creating a peaks over threshold model to capture correlations and clustering of the observational data
      *[Sections 5.1, 5.2, … the same as 4.1, 4.2, … previously]*
6.0   Summary

**EZ MAJOR COMMENT 2** "I wonder why the authors do not employ variograms/semivariances to estimate the spatial correlation time. This will better separate the un-correlated from the correlated variability in their records. In this context I wonder, how much of the increase in correlation times at the hifher aggregation levels may be attributed to well-known scaling effects arising form to the aggregation process itself. The range of variogram estimated from data points increased when these

are aggregated to blocks (or time intervals here) while nugget increases and sill decreases – this is well known from variogram regularization. This effect could be easily determined using data from your Poisson null model."

**Author reply:**

Erwin Zehe has suggested that we use semivariograms as a model for finite range correlations or short-range persistence (i.e., a temporal correlation time), where at a given lag or less in our time series, correlations are observed, but that at that lag or greater, correlations are not observed, or the correlations fall below a statistical threshold. He then further suggests that any increase in the correlation time (based on short-range persistence), might be due to the scaling effects from the aggregation process itself.

We have now done a semivariogram analysis up to long lags on the palaeoflood series, with results shown in a new Appendix B (along with an explanation of what we have done). The results of the semivariogram analysis for long lags (and for the autocorrelation, see reply to Comment #3) is basically that within 95% confidence limits, the signal is overwhelmed by the cyclicity, and we can not conclusively state that there is short-range (vs. long-range) correlations. We cannot therefore rule out the possibility of a power-law decay of the semivariogram with temporal lags. We therefore are unable to calculate a sill.

We also note that in our manuscript after our initial explorations of the time series using autocorrelation, we decided to apply a model of long-range correlations (i.e., one that does not have a temporal correlation time). Certainly, semivariograms can be used a long-range persistence model. However, instead of semivariogram analysis we applied Detrended Fluctuation Analysis (DFA) and Power-Spectral analysis (PS) for quantifying the long-range persistence strength of our palaeoflood and synthetic time series. Both semi-variogram analysis and DFA operate in the time domain, and power-spectral analysis in the frequency domain. In research we did four years ago and published as a review paper (Witt & Malamud, 2013) we examined systematically, using 17,000 synthetic benchmark series, the types of errors when evaluating long-range persistence using these three techniques. All three are appropriate measures for long-range persistence; however, the systematic and random errors (biases and standard errors) of DFA and PS analysis are significantly smaller (particularly for time series with non-Gaussian one-point probability distribution, and with very few values in the time series) compared to semi-variogram analysis, and both DFA and PS analysis are appropriate for a much broader range of long-range persistence strengths compared to semi-variogram analysis. We put our research into the context of the wider literature which has found similar results. We therefore in this paper use DFA and power-spectral analysis, due to it being much more robust than semivariograms as a long-range model. We add to our manuscript a short explanation why we have chosen DFA and power-spectral analysis (compared to semivariograms) for examining long-range persistence.

**EZ MAJOR COMMENT 3** "I appreciate that one prefers the use of power law functions with infinite correlation length – but what is the physical meaning of this?"

**Author reply:** We believe (as this refers to correlation length) this comment refers to our presentation of the auto-correlation function applied to our palaeoflood time series (per year, per decade, per century), and the auto-correlation decaying as an inverse power-law with respect to the temporal lag. We will take "correlation length" here as meaning the lag at which values are separated and still statistically correlated. One could also consider power-spectral analysis and DFA, in terms of an equivalent correlation length for which values are still, on average, statistically correlated. We have added the following to the manuscript "For Eqs. (7) (autocorrelation analysis), (8) (power-spectral analysis), and (12) (DFA), in each case we have an inverse power-law decay with increasing temporal scales (lag, frequency, temporal segment length), which defines a self-affine time series (the time series is statistically self-similar when comparing different temporal scales) (Mandelbrot and Van

Ness, 1968). If the power-law exponent held over 'all' temporal scales (which it rarely does) then the correlation length (the largest lag or temporal scale at which there are still statistical correlations in the time series) would be infinite. One potential significance of this 'infinite' correlation length is that 'all' values are statistical correlated with one another in the time series. A second significance, is that the time series can be explained as a stochastic fractal (Malamud and Turcotte, 1998). In the case of our palaeoflood series this means potentially that the flood timings are organised in time as the points of a Cantor dust (see Ott, 1993 for a discussion on Cantor sets)."

**EZ MINOR COMMENT 1**: "The authors use the presence of gravel enriched strata as indicator for channelized inflow due to a flood into the lake. This seems plausible as those events might be associated with bedload transport and sedimentation into the lake. However, there is no linear relation between flow velocity and bed load transport capacity. The literature is full of empirical formulas for this. This implies that the thickness of these strata is not linearly related to pealk discharge (flow velocities). The authors might consider this as an additional argument against the use of layer thickness as proxy for flood peak discharge."

**Author reply**: We agree that strata (layer) thickness is not an indicator for flood magnitude. Note that in our detrital layers, which represent the flood events, the detrital layers themselves are mainly constituted of dolomite from the catchment, i.e. fine grained sediment with no gravel. We have added words to our original manuscript Data section to clarify this.

"The detritus is mainly constituted of fine, silt-size Triassic dolomite from the catchment (Mangili et al., 2005). Only in the thickest described layers are found fine-sand sized particles in minor amounts. No detrital layers contain gravel. Due to their composition and grain size, these detrital layers can easily be distinguished from the background endogenic sedimentation of the lake. These detrital layers are considered to be the result of channelized streamflow that originated in the hills surrounding the Piànico–Sèllere Basin and triggered by extreme precipitation events (Mangili et al., 2005). The site of the outcrops from which the samples have been taken have been explicitly selected to avoid gravel sediments which could cause hiatuses through erosion. We interpret the site to be a low-energy sedimentary environment in a distal position of the inflowing water."

"Although detrital layers, which are mainly constituted of silt-sized dolomite, indicate the occurrence (season and year) of an extreme precipitation event (flood), the thickness is not taken as representative of the magnitude (e.g., peak discharge or flow velocity) of the event. This is because the thickness of a layer can also be influenced by the distance between the source of the detritus and the studied section and the amount of detritus available at the time of the extreme precipitation and weaker vs. stronger rainfall during the extreme precipitation events (Kämpf et al., 2012, 2015). In addition, over the 9336 yr period we investigate, the source of given rainfall events might result in different thickness signatures (Swierczynski et al., 2012, 2013)."

**EZ MINOR COMMENT 2**: "Why not showing a long time series of annual flood maxima, they also exhibit clustering of extremes and long time periods – this example is maybe closer to the object of desire?"

**Author reply:** We have now upon your suggestion replaced parts C and D of Figure 1 in our manuscript (Atlantic Storm Maximum Wind speed) with flooding events (maximum daily discharge) over 116 years from the Mississippi River, which also clearly illustrates our point of clustering.

**EZ MINOR COMMENT 3**: "Could you also infer on clustering of intensities above a threshold using indicator auto correlations?"

**Author reply:** We understand this question to ask what would happen if we applied the autocorrelation function to our palaeoflood time series (per year, per decade, per century) to all values above a given threshold, in terms of learning more about the clustering (vs. correlation) behaviour of the time series. The values above the threshold could either (time series 1) retain their value, and all values below the threshold set to '0', or (time series 2) all values above the threshold set to a given value (e.g., '1') (and all values below the threshold '0').

Currently we apply autocorrelation analysis to the number of floods per year, per decade, and per century, but without a threshold. The problem we have with our palaeoflood time series, is that rather a time series that is 'just' stochastic, there is also the presence of a long-term cyclicity. In case of weakly-stationary data such as a fractional noise (no underlining long term cycle), we could use the decay rate of the autocorrelation function for quantifying correlations (and also examine values above the threshold to infer clustering behaviour).

However, because of the presence of a long-term cyclicity, the autocorrelation function will be strongly impacted, and we therefore use Detrended Fluctuation Analysis (DFA) and power-spectral analysis, which are more appropriate for fractional noises with cyclicity. For example, DFA is detrending segments of the aggregated time series.

In our manuscript, to examine clustering, we (i) use the one point probability distribution (and in particular the shape parameter of the Weibull distribution) of the time intervals between floods as an indicator for clustering, but also (ii) apply autocorrelation analysis to the interevent occurrence times between events—which is in some ways equivalent to what the reviewer is suggesting (i.e., inferring clustering behaviour based on time between a threshold of 1 flood or greater).

**EZ MINOR COMMENT 4**: "I appreciate the authors effort to provide a reproducible paper, but the explanation of the Poisson process is too detailed, this is text book knowledge. The parameter lambda is the mean value of floods / time interval, which depends on the aggregation level. As far as I know the Poisson process produces cluster data, if lambda is larger than 1?"
**Author reply:**
(a) We have reduced some words and reorganized the section on Poisson processes theory vs. results, but have maintained enough detail that a non-expert will have enough to follow the flow of our manuscript.
(b) The Poisson process as introduced in our original manuscript page 8, does not lead to temporal clustering of the realisation (in time) of a Poisson process, for any lambda > 0 (Cox and Lewis, 1978). We also note that for lambda > 1 the one-point probability distribution of events per time unit is one humped and has a defined mode. We have clarified this in the text.

**EZ MINOR COMMENT 5**: "Page 9 line 5: In case the flood do always occur in the first three years of the decade the average inter event period is not 1 year but, 2/3 *1 + 1/3*7= 3 years?"
**Author reply**: We thank the reviewer for noting this error. We will change

[revised manuscript text omitted]

**Commented [MB2]:** Next two paragraphs were brought in from a later section to here, along with top half of original Figure 3, and histograms to the right.

floods (as from 2 to 4 floods century$^{-1}$ from the 39th to the 39th century) as well as from a few to many  floods (as from 27 to 7 floods century$^{-1}$ from the 464th to the 455th century).

The histograms of these data are also given in **Fig. 3**, on the far right of each time series. The number of years ($k_{year}$) with no floods ($n_{year} = 0$ floods yr$^{-1}$) is $k_{year} = 8530$ yr (92.0% of the record). Similarly, for $n_{year} = 1, 2, 3$ floods yr$^{-1}$, the number of years (respectively) with those values are $k_{year} = 712$ (7.7%), 28 (0.3%), 1 (0.01%) yr. For $N_{year} = 9271$ yr examined, there are 741 yr which have 1 to 3 floods (detrital layers), a total of 771 floods. We will revisit these values in **Sect. 4.1**. The data cover $N_{decade} = 925$ non-overlapping decades (because of the 65 yr gap, eight of the decades within the 9336 yr are disrupted). The number of decades ($k_{decade}$) with no floods ($n_{decade} = 0$ floods decade$^{-1}$) is $k_{decade} = 4$53 decades (49.0% of the record). Similarly, for $n_{decade} = 1, 2, 3$ floods decade$^{-1}$, $k_{decade} = 283$ (30.6%) 123 (13.3%), 43 (4.6%) decades. For $4 \leq n_{decade} \leq 8$ floods decade$^{-1}$, $k_{decade} = 23$ (2.5%) decades. No decades have $n_{decade} > 8$ floods decade$^{-1}$. If we consider the data on the century resolution, we have $N_{century} = 92$ non-overlapping centuries, as the 65 yr gap falls completely within 1 century. The number of centuries ($k_{century}$) with no floods ($n_{century} = 0$ floods century$^{-1}$) is $k_{century} = 1$ century (1% of the record). For $1 \leq n_{century} \leq 5$ and $6 \leq n_{century} \leq 10$ floods century$^{-1}$, $k_{century} = 26$ and 35 centuries (28% and 38% of the record) respectively. For $n_{century} > 10$ floods century$^{-1}$, $k_{century}$ then decreases (**Fig. 3B**) with a maximum of one century that has $n_{century} = 31$ floods century$^{-1}$.

[revised manuscript text omitted]

In this section we willn **Sect. 3.5**, we describe power-spectral analysis and detrended fluctuation analysis as methods for quantifying long-range correlationsstart with a general statistical background on long-range correlations and discuss longshort-range versus long-range correlations, then we describe the idea of long-range persistence and fractional noises. Finally, in **Sect 3.6** we briefly introduce fractional noises.

**,3.1 Interevent occurrence times (IEOTs)**

In this paper, we will consider the number of palaeofloods per year, decade, and century as event series in time. As indicators of clustering and/or correlations (or lack of), we will apply statistical methods to the event magnitudes (i.e., the number of floods per year, decade or century). We will also consider the time intervals between successive events, i.e. the interevent occurrence times (IEOTs) which we introduce here. We will later use the statistical distribution of the IEOTs as an indicator of clustering (or lack of) in the original time series.

[revised manuscript text omitted]

**Commented [MB16]:** Changed in figure:
y-axis for part B. Correct spelling of realization
x-axis, y to yr

[Figure]

**Figure 16: Using the optimized POTFGN+Period model for forecasting the number of floods in one century based on the number of floods in the previous century:** Based on the palaeoflood model realizations for one billion ($10^9$) years are shown **(A)** The 2D probability of the number of floods per century ($x_{century}(j)$) and the number of floods in the succeeding century ($x_{century}(j+1)$) (background grid colours, see legend for scale). The solid grey diagonal line represents equality of the number of floods in both centuries ($x_{century}(j+1) = x_{century}(j)$). **(B)** The two-dimensional probability distribution shown in (A) is cut vertically at $x_{century}$ = 0, 8 and 16 floods century$^{-1}$ and the corresponding probability densities of floods per centuries after a century with $x_{century}$ = 0, 8 and 16 floods century$^{-1}$ are presented as coloured lines (see legend). Also given is the probability density of floods per century of the palaeoflood model.